# Knowing the Unknown: Interpretable Open-World Object Detection via Concept Decomposition Model

Xueqiang Lv [1]   Shizhou Zhang [1]   Yinghui Xing [1]   Di Xu [2]   Peng Wang [1]   Yanning Zhang [1]

## Abstract

Open-world object detection (OWOD) requires incrementally detecting known categories while reliably identifying unknown objects. Existing methods primarily focus on improving unknown recall, yet overlook interpretability, often leading to known–unknown confusion and reduced prediction reliability. This paper aims to make the entire OWOD framework interpretable, enabling the detector to truly "knowing the unknown." To this end, we propose a concept-driven **I**nter**P**retable **OW**OD framework(IPOW) by introducing a Concept Decomposition Model (CDM) for OWOD, which explicitly decomposes the coupled RoI features in Faster R-CNN into discriminative, shared, and background concepts. Discriminative concepts identify the most discriminative features to enlarge the distances between known categories, while shared and background concepts, due to their strong generalization ability, can be readily transferred to detect unknown categories. Leveraging the interpretable framework, we identify that known–unknown confusion arises when unknown objects fall into the discriminative space of known classes. To address this, we propose Concept-Guided Rectification (CGR) to further resolve such confusion. Extensive experiments show that IPOW significantly improves unknown recall while mitigating confusion, and provides concept-level interpretability for both known and unknown predictions.

## 1. Introduction

Traditional object detection approaches are typically built on a closed-set assumption, limiting detection to known

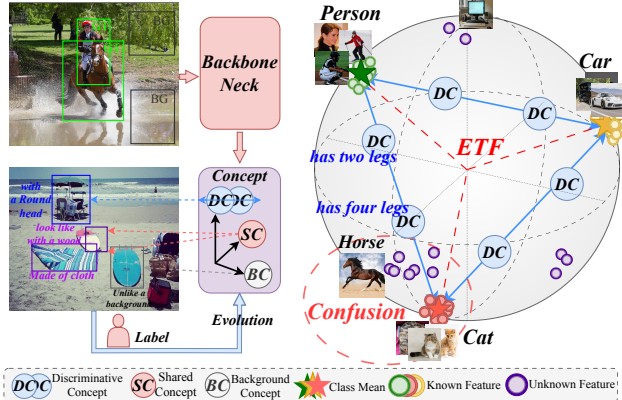

*Figure 1.* **Left:** The proposed IPOW reformulates OWOD as a problem of concept decomposition modeling, consisting of discriminative concepts responsible for known-class recognition, along with shared and background concepts that support unknown object detection. **Right:** Within the IPOW framework, known–unknown confusion can be viewed as unknown objects falling into the discriminative space learned for known classes.

categories that have been seen during training. Open-World Object Detection (OWOD) first introduced by (Joseph et al., 2021a), overcomes this constraint by defining a new framework that can automatically identify previously unseen objects and gradually learn to classify them through subsequent incremental learning stages, thereby enabling continuous adaptation in real-world settings. However, this shift in paradigm brings about two core challenges: *known–unknown confusion*, where visually similar unknown objects are mistakenly detected as known categories, causing a high rate of false positives; and *bias toward known categories*, where detectors trained exclusively on known classes naturally prioritize these known objects, thereby restricting generalization to unknown classes and yielding low recall on unknown objects.

Although there have been recent attempts to tackle these issues, substantial difficulties still persist. Specifically, existing methods (Gupta et al., 2022; Majee et al., 2025) typically employ a class-agnostic objectness head to score all RoIs, heuristically treating high-objectness regions outside known classes as unknown, and predominantly rely on *self-supervised mining based on objectness* to identify latent unknown objects in the training set. Note that the unknown

---

[1]School of Computer Science, Northwestern Polytechnical Unviersity, Xi'an, China [2]Huawei, China. Correspondence to: Shizhou Zhang <szzhang@nwpu.edu.cn>.

*Proceedings of the $43^{rd}$ International Conference on Machine Learning*, Seoul, South Korea. PMLR 306, 2026. Copyright 2026 by the author(s).

classes basically consist of two sources. One is *Known Unknown Classes (KUCs)* (Geng et al., 2020) which denote the unlabeled categories in the training set that cover only a small fraction of the open world. The other is the far more numerous *Unknown Unknown Classes (UUCs)*, which never appear during training, are more common and critical in realistic open-world settings. As self-supervised mining is limited to KUCs, existing methods often misinterpret background regions as latent unknowns, leading to excessive false positives and low unknown recall. Additionally, they represent objectness only in an abstract way and their decision-making procedures do not clarify why certain regions are designated as unknown. More critically, they do not account for the reasons behind confusion between known and unknown classes, which undermines the reliability of their predictions.

In this paper, we employ Concept Bottleneck Models to achieve interpretable knowledge transfer from known to unknown classes, directly addressing the above challenges. To this end, as shown in Fig. 1, we propose a concept-driven **I**nter**P**retable **O**pen-**W**orld object detection framework (**IPOW**) based on a Concept Decomposition Model (CDM) from the perspective of concept modeling and decomposition. Specifically, IPOW is implemented on top of the two-stage detector Faster R-CNN and primarily operates at the RoI head, where each RoI feature is decomposed into three parts: the *discriminative concept*, the *shared concept*, and the *background concept*. Discriminative concepts are used mainly for the classification of known classes, which are devised to capture the most distinctive attributes within known categories to maximize class separation. To enable the detection of unknown categories, a comprehensive shared concept space is constructed by leveraging an LLM to summarize shared semantic attributes across known categories together with learning complementary shared concepts through feature reconstruction. Furthermore, through encoding non-object contextual information, background concepts aim to enhance the detection of potential unknown targets via background inversion, *i.e.* identifying regions that deviate from the surrounding context.

Drawing on the theory of *Neural Collapse* (Papyan et al., 2020), as shown in Fig. 1, known categories collapse into an equiangular tight frame (ETF) structure upon model convergence. This ETF structure captures only the most discriminative features to maximize inter-class separation. Our discriminative concepts explicitly push known categories into this equiangular structure using the most distinctive features. Since discriminative concepts are designed solely for known-class recognition, we observe that unknown objects can also fall into this discriminative space. As shown in Fig. 1 Right, for example, among known classes such as "person" and "cat," the model primarily captures the most distinctive features (two legs vs. four legs) to differ-

entiate them. However, when encountering a four-legged unknown object such as a "horse", the model naturally tends to classify it as a "cat", leading to confusion between known and unknown classes. To resolve this issue, we turn to the shared concept level and propose Concept-Guided Rectification (CGR) based on partial activation of shared concepts. Because shared concepts are mined for known classes, known objects exhibit "full activation" of their corresponding semantic attributes. In contrast, unknown objects only trigger "partial activation" in this shared space. This clear difference in activation patterns allows us to effectively separate unknown objects from known categories. Overall, this concept-based decomposition not only enables effective separation of unknown categories from known ones, but also provides an explanation of *why unknowns are unknown* at the concept level, thereby realizing interpretable knowledge transfer from known to unknown.

Our contributions are summarized as follows:

- We propose a concept-driven interpretable open-world object detection framework by introducing a Concept Decomposition Model to decompose RoI features into discriminative, shared, and background concepts for known and unknown detection.

- Leveraging this interpretable framework, we identify that known–unknown confusion arises from unknown objects falling into the discriminative space of known classes and propose Concept-Guided Rectification (CGR) to mitigate confusion.

- Extensive experiments show that our method achieves state-of-the-art detection performance for known classes and superior recall rate for unknown classes. Moreover, interpretable results for all known and unknown objects can be obtained with the concepts.

## 2. Related Work

**Open-World Object Detection.** OWOD was first formulated in ORE (Joseph et al., 2021a), which augments object detection with contrastive clustering and an energy-based unknown classifier. OW-DETR (Gupta et al., 2022) introduces an attention-driven pseudo-labeling scheme for unknown object discovery, while RandBox (Wang et al., 2023) shows that random region proposals significantly improve recall for unknown objects. OrthogonalDet (Sun et al., 2024) separates objectness detection from classification using orthogonal feature representations, and CROWD (Majee et al., 2025) uses a submodular-function-based strategy for unknown data discovery and representation learning. The work most related to ours is a line of attribute-based methods for open-world detection (Xi et al., 2025a;b), built on pre-trained Open-Vocabulary Object Detectors (OVDs). However, OVD-based methods blur the notion of a truly open

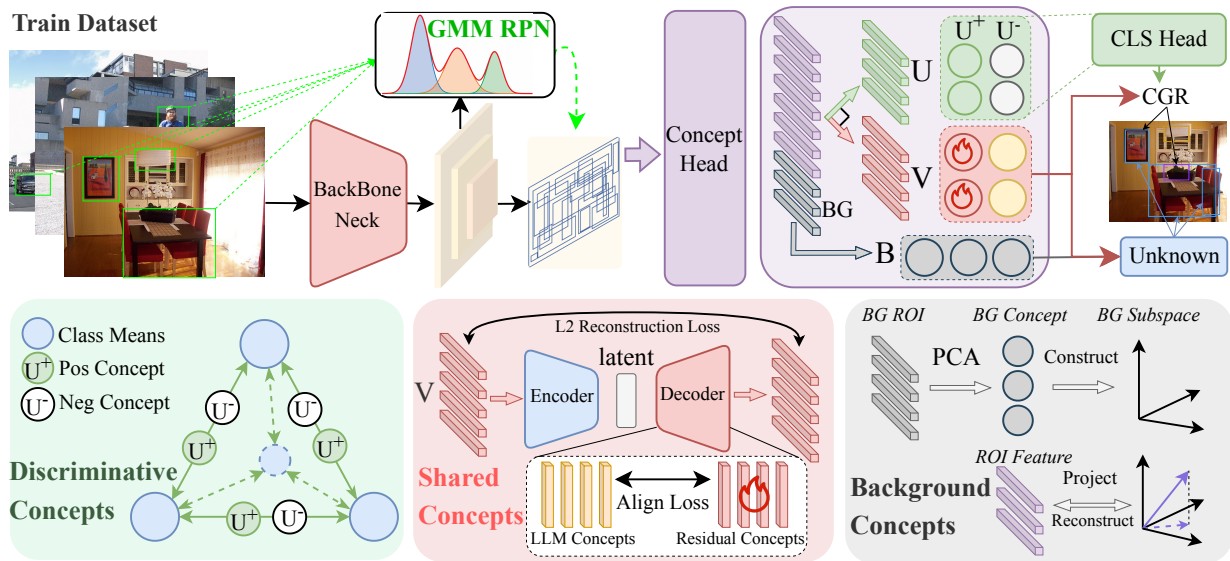

*Figure 2.* Overview of the IPOW framework. GMM-RPN is utilized to mitigate the proposal generation bias towards known-categories. A concept head is adopted before decomposition. Each RoI feature is decomposed into discriminative, shared, and background concepts for known object recognition, transferable unknown discovery, and contextual modeling.

world: trained on massive image–text corpora, they have effectively **"seen"** the unknown classes during pre-training, lacking only explicit labels at inference, which causes implicit data leakage. They also often perform poorly under substantial domain shift from the pre-training data.

**Concept Bottleneck Models.** Concept bottleneck model (CBM) (Koh et al., 2020) introduces an interpretable framework that predicts human-understandable concepts as intermediate representations before making final predictions. (Chauhan et al., 2023) extend CBM by querying human-provided concept labels at inference time to improve prediction accuracy through concept-level interaction. In addition, semi-supervised CBM (Hu et al., 2025) incorporate unlabeled data to improve concept learning with limited annotations, while language-guided CBMs (Yu et al., 2025) leverage natural language supervision to align concept representations with semantic knowledge, further boosting interpretability and generalization. In this paper, we borrow the CBM and propose CDM to reformulate OWOD as an interpretable framework.

## 3. Preliminaries

**Problem Definition.** In line with the conventional OWOD setup, let $\mathcal{T} = \{\mathcal{T}_1, \ldots, \mathcal{T}_n\}$ represent a sequence of tasks arriving over time, each associated with a category set from $\mathcal{C} = \{\mathcal{C}_1, \ldots, \mathcal{C}_n\}$. At any time step $t$, the system maintains a set of cumulative known categories $\mathcal{K}_t = \{\mathcal{C}_1, \ldots, C_t\}$, while the set of unknown classes is explicitly defined as $\mathcal{N}_t = \{C_{t+1}, \ldots\}$, representing categories that may appear in future tasks. The training dataset at time $t$ is denoted as $\mathcal{D}_t = \{(I_i, \mathcal{Y}_i)\}_{i=1}^N$, where $I_i$ denotes the $i_{th}$ image, and

$\mathcal{Y}_i = \{(\mathbf{b}_k, c_k)\}$ denotes its label containing multiple object instances with bounding boxes $\mathbf{b}_k$ and class labels $c_k$. When training task $\mathcal{T}_t$, annotations from previous tasks are unavailable and instances in $\mathcal{N}_t$ are unlabeled. During inference, the model is required to recognize all categories in $\mathcal{K}_t$ and simultaneously identify instances from $\mathcal{N}_t$ as a generic "unknown" class to distinguish them from the background. As it evolves, the model discovers unknown objects and incorporates them into the known set $\mathcal{K}_{t+1}$ once labels become available, enabling an incremental expansion of known categories while mitigating catastrophic forgetting.

**Concept Bottleneck Models.** CBM introduces an interpretable prediction paradigm by explicitly constraining model decisions to pass through a set of human-understandable semantic concepts. Instead of mapping input data to task labels, CBM enforces an intermediate concept space that serves as a bottleneck for reasoning. Formally, given an input $x \in \mathcal{X}$ and label $y \in \mathcal{Y}$, CBM assumes a concept vector $\mathbf{c} \in \mathbb{R}^m$, where each dimension corresponds to a concept, and factorizes prediction into two stages:

$$p(y \mid x) = \sum_c p(y \mid \mathbf{c}) \, p(\mathbf{c} \mid x). \quad (1)$$

Here, $p(\mathbf{c} \mid x)$ models concept prediction, while $p(y \mid \mathbf{c})$ performs task inference solely based on concepts.

## 4. Method

### 4.1. IPOW Framework

The proposed IPOW is built on the decomposition of RoI features together with semantic concept modeling, which

effectively reformulates the OWOD framework for detecting both known and unknown entities. As shown in Fig. 2, IPOW is based on the two-stage detector Faster R-CNN. To eliminate the bias of the RPN towards known categories, we propose a GMM-based RPN for proposal generation; detailed implementation is provided in the Appendix C. Then through a concept head, the concept feature is decomposed and projected into three concept spaces, namely *Discriminative Concepts*, *Shared Concepts* and *Background Concepts*. Discriminative Concepts are responsible for maximizing inter-class margins among known categories, and the resulting concept activation vectors are fed into a classification head for known-class detection. Shared Concepts generalize to unknown object detection through LLM-derived concepts and residual concepts learned via the reconstruction process. Meanwhile, Background Concepts, which model scene context outside object regions, are leveraged to identify regions that are inconsistent with the surrounding background, thereby indicating unknown objects. Finally, both known and unknown predictions are rectified through the Concept-Guided Rectification (CGR) module to alleviate known–unknown confusion.

## 4.2. Concept Decomposition Model

Given an image $I$, each RoI $x_i$ is encoded as

$$\phi(x_i) = \text{RoIAlign}(\text{FPN}(\text{ResNet}(I)), x_i), \quad (2)$$

which is then processed by the Concept Head to produce the concept feature $\mathbf{z}_i \in \mathbb{R}^d$.

$$\mathbf{z}_i = \text{ConceptHead}(\phi(x_i)), \quad (3)$$

where the Concept Head consists of a $1 \times 1$ convolution followed by two linear layers. For simplicity, we omit the subscript $i$ in the following. Since the feature $\mathbf{z}$ captures both the object of interest and the surrounding background context, we decompose it into two distinct vectors, namely the foreground feature $\mathbf{f}_{\text{fg}}$ and the background feature $\mathbf{f}_{\text{bg}}$, which constitute the foreground and background subspaces, respectively:

$$\mathbf{z} = \mathbf{f}_{\text{fg}} + \mathbf{f}_{\text{bg}}. \quad (4)$$

Owing to the mutual exclusivity between foreground and background information, these two components are naturally orthogonal. Subsequently, the extracted foreground feature is projected via orthogonal projections $P_{\mathcal{U}}$ and $P_{\mathcal{V}}$ into two orthogonal vectors:

$$\mathbf{u} = P_{\mathcal{U}}(\mathbf{f}_{\text{fg}}), \qquad \mathbf{v} = P_{\mathcal{V}}(\mathbf{f}_{\text{fg}}). \quad (5)$$

We define the subspace $\mathcal{U}$, in which $\mathbf{u}$ resides, as the *discriminative concept space*, which is utilized to capture the most discriminative features among known classes. In contrast, the subspace $\mathcal{V}$ is defined as the *shared concept space*,

designed to capture semantic attributes shared across known categories, such as having four legs for cats and dogs, or wheels for buses and cars. These shared attributes support generalization for unknown object detection.

The subspaces $\mathcal{U}$ and $\mathcal{V}$ are respectively spanned by discriminative concept vectors $\mathbf{C}^u = \{\mathcal{C}_1^u, \ldots, \mathcal{C}_{K_u}^u\}$ and shared concept vectors $\mathbf{C}^v = \{\mathcal{C}_1^v, \ldots, \mathcal{C}_{K_v}^v\}$. To enforce strict orthogonality, we construct the two subspaces by projecting the original concept vectors using two fixed, mutually orthogonal *orthogonal matrices* $\mathbf{Q}_{\mathcal{U}}$ and $\mathbf{Q}_{\mathcal{V}}$:

$$\mathcal{U} = \text{span}(\mathbf{Q}_{\mathcal{U}} \mathbf{C}^u), \qquad \mathcal{V} = \text{span}(\mathbf{Q}_{\mathcal{V}} \mathbf{C}^v), \quad (6)$$

Accordingly, the foreground space is defined as

$$\mathcal{Z}_{\text{fg}} = \mathcal{U} \oplus \mathcal{V}, \qquad \mathcal{U} \perp \mathcal{V}, \quad (7)$$

and the foreground representation is expressed as

$$\mathbf{f}_{\text{fg}} = \mathbf{u} + \mathbf{v}. \quad (8)$$

By substituting the foreground decomposition, the complete formulation of the concept feature is expressed as:

$$\mathbf{z} = \mathbf{u} + \mathbf{v} + \mathbf{f}_{\text{bg}}. \quad (9)$$

Consequently, an unknown instance can be represented as:

$$\mathbf{z}^{\text{unk}} = \mathbf{u}^{\text{unk}} + \mathbf{v}^{\text{unk}} + \mathbf{f}_{\text{bg}}. \quad (10)$$

In OWOD, unknown instances need not be explicitly classified, so $\mathbf{u}^{\text{unk}}$ is not modeled and the core lies in $\mathbf{v}^{\text{unk}}$. Given the commonalities across object categories, $\mathbf{v}^{\text{unk}}$ and $\mathbf{v}^{\text{known}}$ are expected to be highly overlapping within the shared subspace. Therefore, the model can effectively generalize to unknown categories by modeling the shared concepts derived from known classes. Furthermore, as background encodes only category-agnostic scene context invariant across known and unknown data, explicitly modeling $\mathbf{f}_{\text{bg}}$ facilitates unknown detection by distinguishing foreground from the environment. The details of each concept submodule are as follows.

## 4.3. Discriminative Concepts

According to the Neural Collapse theory (Papyan et al., 2020), when a classifier is trained to convergence on a set of known classes, the feature means of different classes collapse into an *Equiangular Tight Frame* (ETF), referred to as NC2. Formally, let $\{\boldsymbol{\mu}_k\}_{k=1}^K$ denote the class-wise feature means. Neural Collapse implies that

$$\begin{aligned}
\|\boldsymbol{\mu}_k\|_2 &= \|\boldsymbol{\mu}_{k'}\|_2, \\
\langle \boldsymbol{\mu}_k, \boldsymbol{\mu}_{k'} \rangle &= -\frac{1}{K-1}, \quad \forall k \neq k'.
\end{aligned} \quad (11)$$

Motivated by this observation, we define *discriminative concepts* as the most discriminative positive–negative concept

pairs between every two known categories, thereby driving known class representations toward an ETF structure, as illustrated in Fig. 2. Specifically, we leverage an LLM to identify the most discriminative attributes between two classes. Concepts possessing the attribute are treated as positive concepts, while those lacking it are treated as negative concepts, forming a set of discriminative concept pairs $\{(\mathcal{C}_i^{u+}, \mathcal{C}_i^{u-})\}$. Each concept $\mathcal{C}_i^u$ is encoded using the CLIP text encoder to obtain a concept embedding. The activation of concept $\mathcal{C}_i^u$ with respect to the discriminative feature vector $\mathbf{u}$ is computed via cosine similarity:

$$\mathbf{e}_i^u = \text{TextEnc}(\mathcal{C}_i^u), \qquad a(\mathcal{C}_i^u) = \frac{\mathbf{u}^\top \mathbf{e}_i^u}{\|\mathbf{u}\|_2 \|\mathbf{e}_i^u\|_2}. \quad (12)$$

Accordingly, the discriminative concept space is optimized using a margin-based contrastive objective:

$$\mathcal{L}_{\text{disc}} = \sum_i \left[ \|a(\mathcal{C}_i^{u+}) - a(\mathcal{C}_i^{u-})\|_2^2 - \delta \right], \quad (13)$$

where $\delta > 0$ is a margin. Following the standard CBM architecture, we attach a linear classification head to the discriminative concept activations to produce the final classification results :

$$S_{\text{cls}}^k = p(y = k \mid \mathbf{z}) = \text{Softmax}\left(\mathbf{W}_k^\top \mathbf{a}^u\right), \quad (14)$$

where $\mathbf{a}^u = [a(\mathcal{C}_1^u), \ldots, a(\mathcal{C}_{K_u}^u)]^\top$ denotes the vector of activations over all $K_u$ discriminative concepts. such that known-class predictions are determined solely by discriminative concept activations.

### 4.4. Shared Concepts

The shared concept space $\mathcal{V}$ is designed to explicitly construct a comprehensive set of semantic attributes that are shared across known classes. To achieve this, we first summarize shared concepts from known categories using an LLM, producing a set of human-interpretable shared concepts $\{\mathcal{C}_i^v\}_{i=1}^K$. Each shared concept $\mathcal{C}_i^v$ is encoded using the CLIP text encoder, and its activation with respect to the shared feature vector $\mathbf{v}$ is computed via cosine similarity:

$$\mathbf{e}_i^v = \text{TextEnc}(\mathcal{C}_i^v), \quad a(\mathcal{C}_i^v) = \frac{\mathbf{v}^\top \mathbf{e}_i^v}{\|\mathbf{v}\|_2 \|\mathbf{e}_i^v\|_2}. \quad (15)$$

The shared representation optimized via binary cross-entropy supervision, for a given category, the shared concept learning objective is formulated as:

$$\mathcal{L}_{\text{sc}} = -\sum_i \left[ y_i \log(a(\mathcal{C}_i^v)) + (1 - y_i) \log(1 - a(\mathcal{C}_i^v)) \right], \quad (16)$$

where $y_i \in \{0, 1\}$ indicates the presence of the $i$-th shared concept for the category, and $a(\mathcal{C}_i^v)$ denotes the predicted activation of the corresponding shared concept.

However, LLM-derived concepts are inherently incomplete and cannot exhaustively characterize all transferable semantics. To address this limitation, as shown in Fig.2 we employ a sparse auto-encoding mechanism to discover residual shared concepts beyond those summarized by LLM. Given a shared feature $\mathbf{v} \in \mathcal{V}$, we first apply a linear encoder to project it into a low-dimensional sparse activation vector:

$$\alpha = \text{Enc}(\mathbf{v}) = \mathbf{W_e}\mathbf{v}, \qquad \alpha \in \mathbb{R}^m, \quad (17)$$

where $\mathbf{W_e} \in \mathbb{R}^{m \times d}$ and sparsity is enforced on $\alpha$. The decoder consists of a concept dictionary composed of both known and learnable residual shared concept vectors:

$$\mathbf{D} = [\mathbf{D}^{\text{k}}, \ \mathbf{D}^{\text{r}}] \in \mathbb{R}^{d \times m}, \quad (18)$$

where $\mathbf{D}^{\text{k}} = \{\mathbf{e}_i^v\}_{i=1}^K$ corresponds to the LLM-derived shared concepts, and $\mathbf{D}^{\text{r}} = [\mathbf{d}_1^{\text{r}}, \ldots, \mathbf{d}_M^{\text{r}}]$ consists of $M$ randomly initialized, learnable shared concept vectors. The shared feature is reconstructed as

$$\hat{\mathbf{v}} = \mathbf{D}\alpha, \quad \mathcal{L}_{\text{rec}} = \|\mathbf{v} - \hat{\mathbf{v}}\|_2^2, \quad (19)$$

Finally, to ensure that the residual shared concepts are effectively activated and complementary to the LMM-derived ones, we apply the following regularization loss:

$$\mathcal{L}_{\text{align}} = \left\|(\mathbf{D}^{\text{k}})^\top \mathbf{D}^{\text{r}}\right\|_F^2 + \left|\mathcal{E}_{\ell_2}(\mathbf{D}^{\text{k}}) - \mathcal{E}_{\ell_2}(\mathbf{D}^{\text{r}})\right|. \quad (20)$$

Here, $\mathcal{E}_{\ell_2}(\cdot)$ denotes the $\ell_2$ activation energy of a shared concept vector $\mathbf{v}$ over the corresponding concept group.

Accordingly, the LLM-derived and residual shared concepts are unified into a complete shared concept set $\{\mathcal{C}_i^v\}_{i=1}^{K+M}$, with corresponding concept activations $a(\mathcal{C}_i^v)$. The unknowness score is then defined as the maximum activation over the shared concept set: $S_{\text{unk}}^{\text{share}} = \max_i \ a(\mathcal{C}_i^v)$.

### 4.5. Background Concepts

Background Concepts are obtained by applying principal component analysis (PCA) to a set of Background ROI features, producing a set of orthonormal basis vectors. We define these basis vectors as Background Concepts:

$$\mathbf{D}_{\text{bg}} = [\mathcal{C}_1^b, \mathcal{C}_2^b, \ldots, \mathcal{C}_k^b \in \mathbb{R}^{d \times k}], \quad (21)$$

which span the Background Subspace $\mathcal{F}_{\text{bg}} = \text{span}(\mathbf{D}_{\text{bg}})$. Given a RoI Feature $\mathbf{z} \in \mathbb{R}^d$, we compute its reconstruction from the Background Subspace by projecting onto this basis,

$$\hat{\mathbf{z}} = \mathbf{D}_{\text{bg}}\mathbf{D}_{\text{bg}}^\top\mathbf{z}, \quad r(\mathbf{z}) = \|\mathbf{z} - \hat{\mathbf{z}}\|_2 \quad (22)$$

The reconstruction error $r(\mathbf{z})$ measures how well $\mathbf{z}$ can be explained by Background Concepts. A larger error indicates a feature inconsistency with background patterns, i.e., likely belonging to a foreground object, possibly an unknown class. Thus, the foreground score is defined as a normalized reconstruction error, $S_{\text{unk}}^{\text{bg}} = \text{Norm}(r(\mathbf{z}))$ with respect to the feature magnitude, serving as a category-agnostic indicator for unknown object detection.

*Table 1.* OWOD performance comparison against state-of-the-art approaches on M-OWODB (top) and S-OWODB (bottom). Following prior work, duplicate images in the test set are removed in our method. We report U-Recall for unknown classes and mAP for known classes. For fair comparison, * indicates reproduced results using 100 unknown proposals per image. Best results in bold.

| Task IDs(→) | Task 1 | | Task 2 | | | | Task 3 | | | | Task 4 | | |
|---|---|---|---|---|---|---|---|---|---|---|---|---|---|
| | U-Recall | mAP | U-Recall | mAP | | | U-Recall | mAP | | | mAP | | |
| Methods | | Curr. | | Prev. | Curr. | Both | | Prev. | Curr. | Both | Prev. | Curr. | Both |
| **M-OWODB** | | | | | | | | | | | | | |
| ORE (Joseph et al., 2021a) | 4.9 | 56.0 | 2.9 | 52.7 | 26.0 | 39.4 | 3.9 | 38.2 | 12.7 | 29.7 | 29.6 | 12.4 | 25.3 |
| OW-DETR (Gupta et al., 2022) | 7.5 | 59.2 | 6.2 | 53.6 | 33.5 | 42.9 | 5.7 | 38.3 | 15.8 | 30.8 | 31.4 | 17.1 | 27.8 |
| PROB (Zohar et al., 2023) | 19.4 | 59.5 | 17.4 | 55.7 | 32.2 | 44.0 | 19.6 | 43.0 | 22.2 | 36.0 | 35.7 | 18.9 | 31.5 |
| CAT (Ma et al., 2023) | 23.7 | 60.0 | 19.1 | 55.5 | 32.2 | 44.1 | 24.4 | 42.8 | 18.8 | 34.8 | 34.4 | 16.6 | 29.9 |
| RandBox (Wang et al., 2023) | 10.6 | 61.8 | 6.3 | - | - | 45.3 | 7.8 | - | - | 39.4 | - | - | 35.4 |
| OrthogonalDet (Sun et al., 2024) | 24.6 | 61.3 | 26.3 | 55.5 | 38.5 | 47.0 | 29.1 | 46.7 | 30.6 | 41.3 | 42.4 | 24.3 | 37.9 |
| CROWD* (Majee et al., 2025) | 42.9 | 61.7 | 31.4 | 56.7 | 38.9 | 47.8 | 34.7 | 48.0 | 31.4 | 42.5 | 42.9 | 25.4 | 38.5 |
| **IPOW (Ours)** | **50.1** | **62.4** | **41.9** | **61.7** | **43.6** | **52.7** | **46.3** | **49.7** | **35.5** | **45.0** | **46.7** | **28.6** | **42.2** |
| **S-OWODB** | | | | | | | | | | | | | |
| ORE (Joseph et al., 2021a) | 1.5 | 61.4 | 3.9 | 56.7 | 26.1 | 40.6 | 3.6 | 38.7 | 23.7 | 33.7 | 33.6 | 26.3 | 31.8 |
| OW-DETR (Gupta et al., 2022) | 5.7 | 71.5 | 6.2 | 62.8 | 27.5 | 43.8 | 6.9 | 45.2 | 24.9 | 38.5 | 38.2 | 28.1 | 33.1 |
| PROB (Zohar et al., 2023) | 17.6 | 73.4 | 22.3 | 66.3 | 36.0 | 50.4 | 24.8 | 47.8 | 30.4 | 42.0 | 42.6 | 31.7 | 39.9 |
| CAT (Ma et al., 2023) | 24.0 | 74.2 | 23.0 | 67.6 | 35.5 | 50.7 | 24.6 | 51.2 | 32.6 | 45.0 | 45.4 | 35.1 | 42.8 |
| OrthogonalDet (Sun et al., 2024) | 24.6 | 71.6 | 27.9 | 64.0 | 39.9 | 51.3 | 31.9 | 52.1 | 42.2 | 48.8 | 48.7 | 38.8 | 46.2 |
| CROWD* (Majee et al., 2025) | 30.4 | 73.5 | 25.5 | 64.9 | 41.2 | 53.1 | 38.1 | 54.7 | 42.1 | 48.4 | 49.8 | 43.0 | 46.4 |
| **IPOW (Ours)** | **34.7** | **73.6** | **32.6** | **67.9** | **42.3** | **55.1** | **44.3** | **56.3** | **44.6** | **50.5** | **50.2** | **44.6** | **47.4** |

## 4.6. Concept Guided Rectification

Our core insight is that known objects must exhibit full-set activation of their predefined semantic concepts, whereas unknown objects may fall into the discriminative space $\mathcal{U}$ of known categories but typically trigger only partial activation within the shared space $\mathcal{V}$. Based on this insight, we introduce a concept-guided rectification mechanism. For a candidate RoI $x_i$, let $\hat{c}_k \in [0, 1]$ denote the predicted activation of the $k$-th shared concept. For each known class $j$ with its associated shared concept set $\mathcal{C}_j$, and let $|\mathcal{C}_j|$ denote its cardinality. The rectified confidence score $S_{\text{known}}^j$ is defined as:

$$S_{\text{known}}^j = S_{\text{cls}}^j \cdot \left( \prod_{c_k \in \mathcal{C}_j} \hat{c}_k \right)^{\frac{\eta}{|\mathcal{C}_j|}}, \qquad (23)$$

where $S_{\text{cls}}^j$ denotes the raw classification probability obtained by Eqn. 14 and $\eta$ is a scaling factor that controls the strength of concept-guided rectification. Conversely, unknown objects are identified by shared and background concept activations that fail to satisfy the strict criteria of any known class. The rectified unknown score is defined as

$$S_{\text{unk}} = \max\left( S_{\text{unk}}^{\text{share}}, S_{\text{unk}}^{\text{bg}} \right) \cdot \left( 1 - \max_j S_{\text{known}}^j \right), \quad (24)$$

Ultimately, known-category inference is driven by discriminative concepts and rectified through shared concepts, while unknown-category inference is determined by joint activations of shared and background concepts. As a result, all

predictions are grounded in concept-level evidence, leading to robust and interpretable open-world object detection.

## 5. Experiments

### 5.1. Experiment Setup

**Datasets.** We follow the common evaluation protocol (Sun et al., 2024; Wang et al., 2023; Majee et al., 2025; Li et al., 2024; Liu et al., 2024), in which duplicate images in the M-OWODB (Joseph et al., 2021a) test set are removed because the original M-OWODB test set contains 1,127 duplicate images (11.6%). For S-OWODB (Gupta et al., 2022), OW-DETR (Gupta et al., 2022) has already removed duplicate images when constructing the dataset. Details about the dataset can be found in Appendix B.

**Evaluation Metrics.** We evaluate known classes using mean Average Precision (mAP), reported separately for previously seen and newly introduced categories. For unknown object classes, we follow the standard OWOD protocol (Joseph et al., 2021a; Sun et al., 2024) and report Unknown Object Recall (U-Recall). Since U-Recall can be trivially increased by predicting an excessive number of unknown proposals, it may lose its practical significance. We improve the U-Recall metric by limiting the number of post-NMS proposals to at most 100 per image, consistent with the standard Faster R-CNN setting. This refinement enables a fair comparison between different methods in terms of their performance on unknown detection. To quantify confusion between known and unknown classes, we also

*Table 2.* Known–Unknown confusion on M-OWODB. The comparison is shown in terms of U-Recall, WI and A-OSE. Note that these metrics are not calculated for Task 4 because all 80 classes are known. Best results in bold.

| Task IDs (→) | Task 1 | | | Task 2 | | | Task 3 | | |
|---|---|---|---|---|---|---|---|---|---|
| Method | U-Recall (↑) | WI (↓) | A-OSE (↓) | U-Recall (↑) | WI (↓) | A-OSE (↓) | U-Recall (↑) | WI (↓) | A-OSE (↓) |
| ORE (Joseph et al., 2021a) | 4.9 | 0.0621 | 10459 | 2.9 | 0.0282 | 10445 | 3.9 | 0.0211 | 7990 |
| OW-DETR (Gupta et al., 2022) | 7.5 | 0.0571 | 10240 | 6.2 | 0.0278 | 8441 | 5.7 | 0.0156 | 6803 |
| PROB (Zohar et al., 2023) | 19.4 | 0.0569 | 5195 | 17.4 | 0.0344 | 6452 | 19.6 | 0.0151 | 2641 |
| RandBox (Wang et al., 2023) | 10.6 | **0.0240** | 4498 | 6.3 | **0.0078** | 1880 | 7.8 | **0.0054** | 1452 |
| OrthogonalDet (Sun et al., 2024) | 24.6 | 0.0299 | 4148 | 26.3 | 0.0099 | 1791 | 29.1 | 0.0077 | 1345 |
| CROWD* (Majee et al., 2025) | 42.9 | 0.0380 | 3823 | 31.4 | 0.0101 | 1508 | 34.7 | 0.0066 | 1266 |
| **Ours** | **50.1** | 0.0369 | **3648** | **41.9** | 0.0098 | **1460** | **46.3** | 0.0062 | **1126** |

*Table 3.* The ablation study of each component on the M-OWODB benchmark across Tasks 1–3. Best results in bold.

| Task IDs (→) | Task 1 | | | | Task 2 | | | | Task 3 | | | |
|---|---|---|---|---|---|---|---|---|---|---|---|---|
| Methods | U-Recall (↑) | mAP (↑) | WI (↓) | A-OSE (↓) | U-Recall (↑) | mAP (↑) | WI (↓) | A-OSE (↓) | U-Recall (↑) | mAP (↑) | WI (↓) | A-OSE (↓) |
| Base Model | 21.8 | 61.1 | 0.0566 | 6765 | 14.7 | 51.9 | 0.0268 | 2543 | 17.7 | 43.8 | 0.0151 | 1676 |
| + GMM RPN | 25.9 | 61.7 | 0.0536 | 8029 | 19.1 | 52.1 | 0.0269 | 3897 | 22.4 | 43.6 | 0.0164 | 2670 |
| + Discriminative Concepts | 23.3 | 62.4 | 0.0492 | 6778 | 19.3 | 52.4 | 0.0273 | 3721 | 21.4 | 44.2 | 0.0173 | 2720 |
| + Share Concept | 45.4 | 62.6 | 0.0460 | 6264 | 37.6 | 52.5 | 0.0286 | 3746 | 40.5 | 44.7 | 0.0156 | 2650 |
| + BG Concept | **50.4** | 62.6 | 0.0460 | 6264 | **42.4** | 52.5 | 0.0286 | 3746 | **46.9** | 44.7 | 0.0156 | 2650 |
| + Concept-Guided Rectification | 50.1 | 62.4 | **0.0369** | **3648** | 41.9 | **52.7** | **0.0098** | **1460** | 46.3 | **45.0** | **0.0062** | **1126** |

*Table 4.* OWOD results on the Remote Sensing DIOR dataset.

| Task IDs | Task 1 | | | | Task 2 | | |
|---|---|---|---|---|---|---|---|
| Methods | U-Recall | mAP Curr. | WI | A-OSE | mAP Prev. | Curr. | Both |
| Base Model | 2.8 | 69.8 | 0.0743 | 1833 | 67.0 | 64.3 | 65.6 |
| IPOW | **19.9** | **71.2** | **0.0573** | **1143** | **68.2** | **66.3** | **67.2** |

report Wilderness Impact (WI) (Dhamija et al., 2020) and Absolute Open-Set Error (A-OSE) (Miller et al., 2018).

**Implementation Details.** Details about the implementation can be found in Appendix B.

## 5.2. Performance on OWOD

**Results on M-OWODB.** As shown in the upper part of Table 1, IPOW achieves the best overall performance on M-OWODB. Compared to previous state-of-the-art method CROWD, IPOW improves U-Recall by *7.2*, *10.5*, and *11.6* points on Tasks 1–3, respectively, achieving remarkable U-Recall rates of 50.1%, 41.9% and 46.3%. Meanwhile, IPOW also achieves the best known-class mAP across tasks, with mAP values of 62.4, 52.7, 45.0, and 42.2 on both classes. As more known categories are introduced, IPOW achieves larger U-Recall gains over existing methods, suggesting increasingly rich and transferable concepts consistent with open-world detection practical settings.

**Results on S-OWODB.** On the superclass-separated S-OWODB benchmark, IPOW consistently achieves the best

performances, as shown in the lower part of Table 1. Compared to CROWD, IPOW improves U-Recall by *4.3*, *7.1*, and *6.2* points in Tasks 1–3, respectively, achieving the highest U-Recall in all tasks (34.7%, 32.6% and 44.3%). Meanwhile, IPOW also consistently improves the mAP of known categories, reaching 73.6, 55.1, 50.5, and 47.4 on Tasks 1–4, respectively. Note that S-OWODB is more challenging for our IPOW, as superclass-level separation reduces the number of transferable shared concepts. Despite this difficulty, IPOW can still significantly outperform previous methods, showing that concept decomposition model is an effective solution for OWOD.

**Known–Unknown Confusion Analysis.** We further analyze known–unknown confusion in Table 2. Across all tasks, IPOW consistently achieves the lowest A-OSE, reducing false known predictions to 3648, 1460, and 1126, respectively. Compared to CROWD, this corresponds to reductions of 175, 48, and 140 unknown objects being wrongly classified as any of the known class. Meanwhile, IPOW maintains highly competitive Wilderness Impact (WI) values of 0.0369, 0.0098, and 0.0062. These results clearly show that our method has a great advantage in reducing known-unknown confusion and the proposed concept-guided rectification mechanism is effective in mitigating such confusion.

**Results on the Remote Sensing Dataset DIOR.** As shown in Table 4, IPOW significantly improves U-Recall on DIOR, increasing U-Recall from 2.8% to 19.9% on Task 1, and greatly reduces known–unknown confusion, decreasing WI from 0.0743 to 0.0573 and A-OSE from 1833 to 1143. In

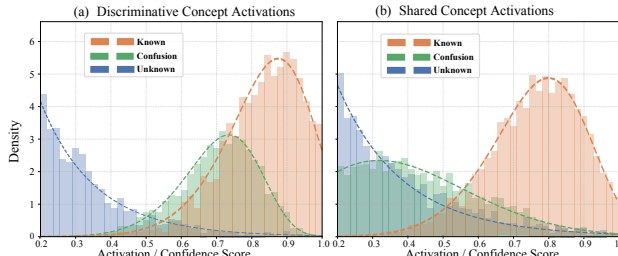

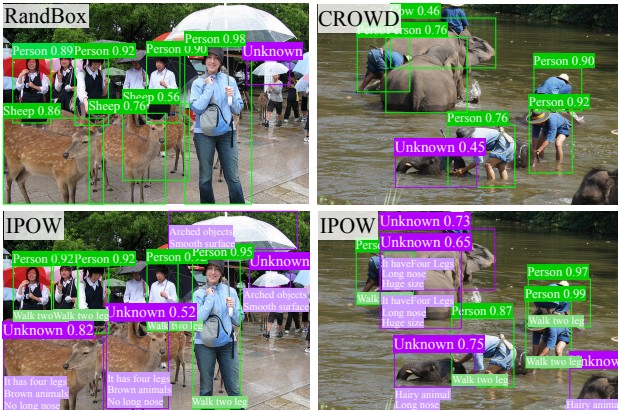

*Figure 3.* Score distributions on M-OWODB Task 1 for known, confusion (unknown falsely predicted as known), and unknown samples. Discriminative concept activations (left) and the geometric mean of class-specific shared concept activations (right).

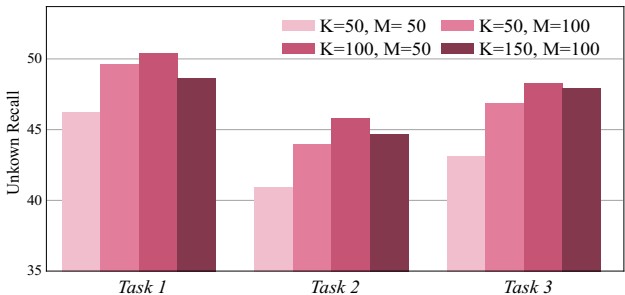

*Figure 4.* Ablation study on the number of shared concepts on M-OWODB. $K$ and $M$ denote the numbers of LLM-derived shared concepts and residual shared concepts, respectively.

*Figure 5.* Qualitative results contrasting IPOW with RandBox, and CROWD, demonstrating that IPOW offers interpretable concept-level reasoning, reduces known–unknown confusion, and enables effective generalization to unknown objects.

(unknown falsely predicted as known categories) exhibit highly activated discriminative concepts, resembling known objects in the discriminative space. However, as illustrated in the shared concept group on the right, these confused unknown instances can be explicitly excluded.

**Number of Shared Concepts.** As shown in Fig. 4, we vary the numbers of LLM-derived and residual shared concepts. The best performance is achieved with $K = 100$ and $M = 50$, demonstrating that residual shared concepts effectively complement LLM-derived ones. In contrast, increasing the numbers to $K = 150$ and $M = 100$ degrades recall, indicating that excessive shared concepts introduce redundancy and harm unknown detection.

### 5.4. Qualitative Analysis

As shown in Fig. 5, IPOW provides a concept-level explanation for all known and unknown predictions with activated concepts. For unknown objects, IPOW allows users to clearly localize objects of interest and identify relevant semantics. This interpretability facilitates user annotation of unknown objects and their incorporation into subsequent tasks for incremental learning. Compared with existing methods, IPOW shows higher unknown recall while effectively mitigating known–unknown confusion.

## 6. Conclusions

In this paper, we proposed IPOW, a concept-driven interpretable framework for open-world object detection. Inspired by Concept Bottleneck Models, IPOW introduces a Concept Decomposition Model (CDM) that decomposes RoI features into discriminative, shared, and background concepts, enabling structured and interpretable reasoning over both known and unknown objects. Through this formulation, we reveal that known–unknown confusion arises

addition, for known classes, IPOW improves mAP by 1.4 and 1.6 points in Task 1–2, respectively. These results show that IPOW generalizes well to remote sensing scenarios that differ significantly from everyday natural images.

### 5.3. Ablation Study

**Effectiveness of Main Components.** Table 3 reports the contribution of each component of IPOW on M-OWODB. Adding the GMM-based RPN improves unknown recall by 4.1–4.7 points across Tasks 1–3 by reducing the RPN's bias toward known categories. Discriminative concepts benefit known-class recognition, improving known-class mAP by 0.3–0.7 points, indicating their role in strengthening known class separation. In contrast, introducing shared concepts yields substantial gains in U-Recall (18.3–22.1 points), demonstrating that shared semantic concepts are the key factors for generalizing knowledge to unseen categories. Further incorporating background concepts provides additional U-Recall improvements (4.8–6.4 points) by explicitly modeling non-object semantics, which helps distinguish unknown objects from background clutter. Finally, CGR significantly reduces known–unknown confusion, achieving a reduction of 19.7%, 65.7% and 60.2% in WI and 41.7%, 61.0% and 57.5% in A-OSE across 3 tasks, highlighting its critical role in resolving known–unknown confusion.

**Statistical Analysis of Known-Unknown Confusion.** As shown in Fig. 3 (Left), those confused unknown objects

when unknown objects fall into the discriminative space learned for known classes. To address this issue, we further propose Concept-Guided Rectification (CGR), which leverages shared concept activations to effectively suppress such confusion in a principled and interpretable manner. Overall, IPOW demonstrates that concept-level decomposition provides a powerful solution for interpretable knowledge transfer in open-world detection, offering both improved reliability and transparent decision-making.

## Acknowledgements

This work was supported in part by the National Natural Science Foundation of China (NSFC) under Grant 62576282, 62476223.

## Impact Statement

This paper presents work whose goal is to advance the field of machine learning. There are many potential societal consequences of our work, none of which we feel must be specifically highlighted here.

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

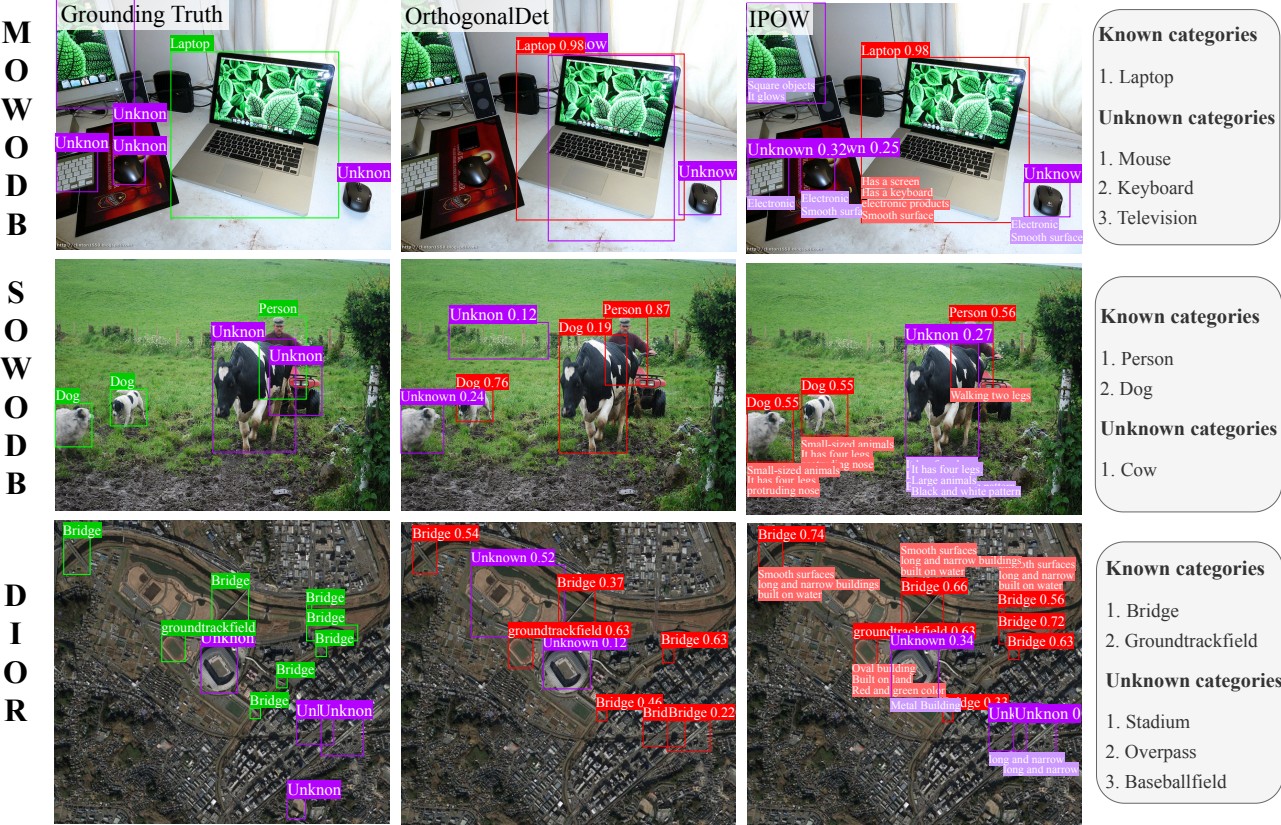

*Figure 6.* Qualitative results comparing IPOW with OrthogonalDet on M-OWODB, S-OWODB, and the DIOR dataset, showing that IPOW provides interpretable concept-level reasoning and more effectively reduces known–unknown confusion across diverse scenarios.

## A. Visualization

As shown in Fig. 6, we qualitatively compare IPOW with OrthogonalDet on M-OWODB, S-OWODB, and the remote sensing DIOR dataset. The results highlight the interpretability of IPOW, where both known and unknown predictions are explained through activated semantic concepts.

IPOW effectively mitigates known–unknown confusion. For instance, in S-OWODB, visually similar objects such as *cow* are misclassified as the known class *dog* by OrthogonalDet, whereas IPOW excludes such instances from known categories based on incomplete shared concept activations, correctly identifying them as unknown. This provides a clear concept-level explanation of why these instances should not be classified as known.

Furthermore, results on DIOR show that IPOW remains effective in remote sensing scenarios with distributions markedly different from natural images, producing reliable unknown predictions with interpretable concept responses.

## B. Experiment setup details

**Datasets.** We follow the common evaluation protocol (Sun et al., 2024; Wang et al., 2023) and evaluate all methods on both open-world and incremental object detection benchmarks. For open-world object detection, we consider the superclass-mixed benchmark M-OWODB (Joseph et al., 2021a) and the superclass-separated benchmark S-OWODB (Gupta et al., 2022). M-OWODB combines PASCAL VOC and COCO, where all VOC categories are treated as known classes in Task 1, and the remaining COCO categories are introduced incrementally as unknown classes. In contrast, S-OWODB uses only the COCO dataset and strictly separates superclasses across tasks, resulting in more isolated semantic partitions and a smaller amount of training data compared to M-OWODB. For incremental object detection, we adopt the class splits of VOC 2007 proposed in (Shmelkov et al., 2017), which include multiple two-stage incremental settings (10+10, 15+5, and 19+1).

To further investigate whether IPOW remains effective under scenarios with significantly different data distributions from common daily scenes such as COCO and PASCAL VOC, we conduct open-world experiments on the popular remote sensing

*Table 5.* Comparison with state-of-the-art methods in single-step incremental object detection settings on PASCAL VOC 2007. Best results are shown in **bold**, while newly introduced classes in each task are shaded in gray. Our method consistently outperforms existing approaches across all settings.

| 10 + 10 setting | aero | cycle | bird | boat | bottle | bus | car | cat | chair | cow | table | dog | horse | bike | person | plant | sheep | sofa | train | tv | mAP |
|---|---|---|---|---|---|---|---|---|---|---|---|---|---|---|---|---|---|---|---|---|---|
| ILOD (Shmelkov et al., 2017) | 69.9 | 70.4 | 69.4 | 54.3 | 48 | 68.7 | 78.9 | 68.4 | 45.5 | 58.1 | 59.7 | 72.7 | 73.5 | 73.2 | 66.3 | 29.5 | 63.4 | 61.6 | 69.3 | 62.2 | 63.2 |
| Faster ILOD (Peng et al., 2020) | 72.8 | 75.7 | 71.2 | 60.5 | 61.7 | 70.4 | 83.3 | 76.6 | 53.1 | 72.3 | 36.7 | 70.9 | 66.8 | 67.6 | 66.1 | 24.7 | 63.1 | 48.1 | 57.1 | 43.6 | 62.1 |
| ORE (Joseph et al., 2021a) | 63.5 | 70.9 | 58.9 | 42.9 | 34.1 | 76.2 | 80.7 | 76.3 | 34.1 | 66.1 | 56.1 | 70.4 | 80.2 | 72.3 | 81.8 | 42.7 | 71.6 | 68.1 | 77.0 | 67.7 | 64.5 |
| Meta-ILOD (Joseph et al., 2021b) | 76.0 | 74.6 | 67.5 | 55.9 | 57.6 | 75.1 | 85.4 | 77.0 | 43.7 | 70.8 | 60.1 | 66.4 | 76.0 | 72.6 | 74.6 | 39.7 | 64.0 | 60.2 | 68.5 | 60.7 | 66.3 |
| ROSETTA (Yang et al., 2022) | 74.2 | 76.2 | 64.9 | 54.4 | 57.4 | 76.1 | 84.4 | 68.8 | 52.4 | 67.0 | 62.9 | 63.3 | 79.8 | 72.8 | 78.1 | 40.1 | 62.3 | 61.2 | 72.4 | 66.8 | 66.8 |
| OW-DETR(Gupta et al., 2022) | 61.8 | 69.1 | 67.8 | 45.8 | 47.3 | 78.3 | 78.4 | 78.6 | 36.2 | 71.5 | 57.5 | 75.3 | 76.2 | 77.4 | 79.5 | 40.1 | 66.8 | 66.3 | 75.6 | 64.1 | 65.7 |
| PROB (Zohar et al., 2023) | 70.4 | 75.4 | 67.3 | 48.1 | 55.9 | 73.5 | 78.5 | 75.4 | 42.8 | 72.2 | 64.2 | 73.8 | 76.0 | 74.8 | 75.3 | 40.2 | 66.2 | 73.3 | 64.4 | 64.0 | 66.5 |
| CAT (Ma et al., 2023) | 76.5 | 75.7 | 67.0 | 51.0 | 62.4 | 73.2 | 82.3 | 83.7 | 42.7 | 64.4 | 56.8 | 74.1 | 75.8 | 79.2 | 78.1 | 39.9 | 65.1 | 59.6 | 78.4 | 67.4 | 67.7 |
| OrthogonalDet (Sun et al., 2024)[1] | 82.9 | 80.1 | 75.8 | 64.3 | 60.6 | 81.5 | 87.9 | 54.9 | 48 | 82.1 | 57.7 | 63.5 | 80.5 | 77.6 | 78.2 | 38.9 | 69.8 | 62.8 | 76.9 | 64.2 | 69.41 |
| CROWD (Majee et al., 2025) | 84.1 | 84.5 | 73.9 | 60.0 | 65.1 | 80.1 | 89.3 | 82.7 | 53.3 | 77.4 | 63.4 | 78.5 | 80.9 | 83.4 | 83.9 | 46.5 | 72.6 | 60.9 | 77.9 | 71.5 | 73.5 |
| **IPOW (ours)** | 79.8 | 83.7 | 78.0 | 63.1 | 65.1 | 81.3 | 87.0 | 84.8 | 62.0 | 71.3 | 64.9 | 82.5 | 79.0 | 79.1 | 83.8 | 48.6 | 66.0 | 67.8 | 74.9 | 74.2 | **73.8** |

| 15 + 5 setting | aero | cycle | bird | boat | bottle | bus | car | cat | chair | cow | table | dog | horse | bike | person | plant | sheep | sofa | train | tv | mAP |
|---|---|---|---|---|---|---|---|---|---|---|---|---|---|---|---|---|---|---|---|---|---|
| ILOD (Shmelkov et al., 2017) | 70.5 | 79.2 | 68.8 | 59.1 | 53.2 | 75.4 | 79.4 | 78.8 | 46.6 | 59.4 | 59.0 | 75.8 | 71.8 | 78.6 | 69.6 | 33.7 | 61.5 | 63.1 | 71.7 | 62.2 | 65.8 |
| Faster ILOD (Peng et al., 2020) | 66.5 | 78.1 | 71.8 | 54.6 | 61.4 | 68.4 | 82.6 | 82.7 | 52.1 | 74.3 | 63.1 | 78.6 | 80.5 | 78.4 | 80.4 | 36.7 | 61.7 | 59.3 | 67.9 | 59.1 | 67.9 |
| ORE (Joseph et al., 2021a) | 75.4 | 81.0 | 67.1 | 51.9 | 55.7 | 77.2 | 85.6 | 81.7 | 46.1 | 76.2 | 55.4 | 76.7 | 86.2 | 78.5 | 82.1 | 32.8 | 63.6 | 54.7 | 77.7 | 64.6 | 68.5 |
| Meta-ILOD (Joseph et al., 2021b) | 78.4 | 79.7 | 66.9 | 54.8 | 56.2 | 77.7 | 84.6 | 79.1 | 47.7 | 75.0 | 61.8 | 74.7 | 81.6 | 77.5 | 80.2 | 37.8 | 58.0 | 54.6 | 73.0 | 56.1 | 67.8 |
| ROSETTA (Yang et al., 2022) | 76.5 | 77.5 | 65.1 | 56.0 | 60.0 | 78.3 | 85.5 | 78.7 | 49.5 | 68.2 | 67.4 | 71.2 | 83.9 | 75.7 | 82.0 | 43.0 | 60.6 | 64.1 | 72.8 | 67.4 | 69.2 |
| OW-DETR (Gupta et al., 2022) | 77.1 | 76.5 | 69.2 | 51.3 | 61.3 | 79.8 | 84.2 | 81.0 | 49.7 | 79.6 | 58.1 | 79.0 | 83.1 | 67.8 | 85.4 | 33.2 | 65.1 | 62.0 | 73.9 | 65.0 | 69.4 |
| PROB (Zohar et al., 2023) | 77.9 | 77.0 | 77.5 | 56.7 | 63.9 | 75.0 | 85.5 | 82.3 | 50.0 | 78.5 | 63.1 | 75.8 | 80.0 | 78.3 | 77.2 | 38.4 | 69.8 | 57.1 | 73.7 | 64.9 | 70.1 |
| CAT (Ma et al., 2023) | 75.3 | 81.0 | 84.4 | 64.5 | 56.6 | 74.4 | 84.1 | 86.6 | 53.0 | 70.1 | 72.4 | 83.4 | 85.5 | 81.6 | 81.0 | 32.0 | 58.6 | 60.7 | 81.6 | 63.5 | 72.2 |
| OrthogonalDet (Sun et al., 2024)[1] | 81.8 | 79.3 | 71.0 | 71.0 | 58.8 | 62.1 | 82.6 | 89.7 | 79.8 | 47.0 | 80.5 | 61.1 | 79.9 | 80.2 | 81.6 | 44.2 | 65.5 | 71.5 | 75.6 | 74.2 | 72.6 |
| CROWD(Majee et al., 2025) | 82.8 | 80.6 | 72.5 | 59.6 | 61.3 | 83.1 | 89.3 | 83 | 49.2 | 86.1 | 62.2 | 83.7 | 86 | 80.3 | 82.8 | 46.1 | 80 | 63.7 | 79.5 | 75.6 | 74.4 |
| **IPOW (ours)** | 77.9 | 79.5 | 77.0 | 65.7 | 63.0 | 82.2 | 86.2 | 87.7 | 56.6 | 82.8 | 68.9 | 86.7 | 85.9 | 79.6 | 84.8 | 44.6 | 71.7 | 68.6 | 74.6 | 75.2 | **75.0** |

| 19 + 1 setting | aero | cycle | bird | boat | bottle | bus | car | cat | chair | cow | table | dog | horse | bike | person | plant | sheep | sofa | train | tv | mAP |
|---|---|---|---|---|---|---|---|---|---|---|---|---|---|---|---|---|---|---|---|---|---|
| ILOD (Shmelkov et al., 2017) | 69.4 | 79.3 | 69.5 | 57.4 | 45.4 | 78.4 | 79.1 | 80.5 | 45.7 | 76.3 | 64.8 | 77.2 | 80.8 | 77.5 | 70.1 | 42.3 | 67.5 | 64.4 | 76.7 | 62.7 | 68.2 |
| Faster ILOD (Peng et al., 2020) | 64.2 | 74.7 | 73.2 | 55.5 | 53.7 | 70.8 | 82.9 | 82.6 | 51.6 | 79.7 | 58.7 | 78.8 | 81.8 | 75.3 | 77.4 | 43.1 | 73.8 | 61.7 | 69.8 | 61.1 | 68.5 |
| ORE (Joseph et al., 2021a) | 67.3 | 76.8 | 60 | 48.4 | 58.8 | 81.1 | 86.5 | 75.8 | 41.5 | 79.6 | 54.6 | 72.8 | 85.9 | 81.7 | 82.4 | 44.8 | 75.8 | 68.2 | 75.7 | 60.1 | 68.8 |
| Meta-ILOD (Joseph et al., 2021b) | 78.2 | 77.5 | 69.4 | 55.0 | 56.0 | 78.4 | 84.2 | 79.2 | 46.6 | 79.0 | 63.2 | 78.5 | 82.7 | 79.1 | 79.9 | 44.1 | 73.2 | 66.3 | 76.4 | 57.6 | 70.2 |
| ROSETTA (Yang et al., 2022) | 75.3 | 77.9 | 65.3 | 56.2 | 55.3 | 79.6 | 84.6 | 72.9 | 49.2 | 73.7 | 68.3 | 71.0 | 78.9 | 77.7 | 80.7 | 44.0 | 69.6 | 68.5 | 76.1 | 68.3 | 69.6 |
| OW-DETR (Gupta et al., 2022) | 70.5 | 77.2 | 73.8 | 54.0 | 55.6 | 79.0 | 80.8 | 80.6 | 43.2 | 80.4 | 53.5 | 77.5 | 89.5 | 82.0 | 74.7 | 43.3 | 71.9 | 66.6 | 79.4 | 62.0 | 70.2 |
| PROB (Zohar et al., 2023) | 80.3 | 78.9 | 77.6 | 59.7 | 63.7 | 75.2 | 86.0 | 83.9 | 53.7 | 82.8 | 66.5 | 82.7 | 80.6 | 83.8 | 77.9 | 48.9 | 74.5 | 69.9 | 77.6 | 48.5 | 72.6 |
| CAT (Ma et al., 2023) | 86.0 | 85.8 | 78.8 | 65.3 | 61.3 | 71.4 | 84.8 | 84.8 | 52.9 | 78.4 | 71.6 | 82.7 | 83.8 | 81.2 | 80.7 | 43.7 | 75.9 | 58.5 | 85.2 | 61.1 | 73.8 |
| OrthogonalDet (Sun et al., 2024)[1] | 81.8 | 82.6 | 77.0 | 56.3 | 66.0 | 74.4 | 88.5 | 78.7 | 51.2 | 84.3 | 63.1 | 84.4 | 81.3 | 78.8 | 80.9 | 46.8 | 77.9 | 68.6 | 74.1 | 74.5 | 73.6 |
| CROWD (Majee et al., 2025) | 81.7 | 80.3 | 77.4 | 57.2 | 66.8 | 80.7 | 87.1 | 67.9 | 49.4 | 87.3 | 65.6 | 84.2 | 85.4 | 79.9 | 81.6 | 48.6 | 77.0 | 69.0 | 82.2 | 75.3 | 74.2 |
| **IPOW (ours)** | 81.1 | 78.7 | 78.2 | 56.3 | 63.5 | 77.2 | 86.2 | 86.8 | 59.5 | 79.8 | 64.7 | 86.3 | 83.0 | 78.8 | 83.2 | 49.3 | 74.4 | 71.6 | 81.0 | 71.0 | **74.5** |

dataset DOIR (Li et al., 2020). DOIR is a remote sensing object detection benchmark consisting of 20 object categories with diverse aerial viewpoints. We partition the dataset into two tasks according to the alphabetical order of category names, with 10 classes in each task. During training and evaluation of the first task, the classes belonging to the second task are treated as unknown categories.

**Implementation Details.** Our method is built upon Faster R-CNN (Ren et al., 2016) and employs a ResNet-50 (He et al., 2016) backbone pretrained on ImageNet, without building upon any existing OWOD methods. The model is trained using the SGD optimizer with a learning rate of 0.02 and a batch size of 16. For the shared concept space, the numbers of LLM-derived and residual shared concepts are set to $K = 100$ and $M = 50$, respectively. In the concept-guided rectification module, the rectification strength parameter $\eta$ is set to 0.8. Note that in our method, the LLM is used only for semantic modeling when new tasks are introduced and does not participate in training or inference, incurring no additional overhead.

## C. GMM RPN

In Faster R-CNN, the RPN generates candidates but suffers from a strong bias toward known categories, as unannotated unknown objects are often suppressed as background. While sampling proposals from random Gaussian noise (Wang et al., 2023) can mitigate this bias, it sacrifices the RPN's refinement capability and necessitates multiple forward passes, significantly increasing inference overhead.

We propose an efficient GMM-based strategy based on the observation that objects, regardless of category, exhibit consistent spatial and scale priors: they are more frequently located toward the center of the image and occupy medium-sized bounding

boxes. This pattern naturally aligns with a Gaussian distribution. Thus, we employ a Gaussian Mixture Model (GMM) to fit the distribution of known ground-truth boxes, capturing universal objectness priors that generalize to the open world without additional cost. Formally, we represent each bounding box as $\mathbf{b} = [x, y, w, h]$ and model the spatial-scale distribution of known categories using a Gaussian Mixture Model (GMM):

$$P(\mathbf{b}) = \sum_{k=1}^{K} \pi_k \mathcal{N}(\mathbf{b}|\boldsymbol{\mu}_k, \boldsymbol{\Sigma}_k), \tag{25}$$

where $\pi_k$, $\boldsymbol{\mu}_k$, and $\boldsymbol{\Sigma}_k$ denote the mixing weights, means, and covariances of the $k$-th Gaussian component, respectively. By modeling intrinsic geometric priors, the GMM captures universal object distributions that mitigate known-category bias. We retain the original learnable RPN for precise known-class detection and use GMM-sampled proposals as a zero-cost complement, providing spatially informed candidates for both known and unknown objects without additional inference overhead.

## D. Incremental Object Detection

Incremental Object Detection (IOD), as a subtask of OWOD, aims to incrementally acquire new object categories over time while retaining the ability to detect previously learned ones. In this way, OWOD is able to discover categories of interest from unknown objects and learn them progressively, enabling the detector to continuously evolve in real-world scenarios.

Previous works following OW-DETR (Gupta et al., 2022) employ exemplar replay-based fine-tuning to alleviate catastrophic forgetting. However, as noted in (Zhang et al., 2025), catastrophic forgetting in incremental object detection stems from a more fundamental issue, namely *foreground–background confusion*, where unannotated objects from previously learned classes are misclassified as background during training. Exemplar replay alone cannot effectively address this problem and also incurs additional memory and computational overhead. Instead, following (Zhang et al., 2025), we adopt a pseudo-labeling strategy that leverages models from previous tasks to generate pseudo labels for previously learned classes, thereby mitigating foreground–background confusion during incremental learning.

## E. IOD Benchmark Results

Explicitly identifying unknown objects transforms the abrupt introduction of new categories into a progressive learning process from previously discovered unknown instances, enabling better knowledge preservation during adaptation to new tasks. In IPOW, strong unknown detection capability is achieved, while known categories are detected through discriminative concepts organized into an Equiangular Tight Frame (ETF) structure. During incremental learning, new categories are separated by additional discriminative concepts without altering the existing discriminative space, allowing most previously acquired knowledge to be preserved.

As shown in Table 5, IPOW consistently outperforms existing methods across all single-step incremental settings. Specifically, IPOW surpasses CROWD by 0.3, 0.6, and 0.3 mAP under the 10+10, 15+5, and 19+1 settings, achieving mAP scores of 73.8, 75.0, and 74.5, respectively. These results demonstrate that the proposed concept-based incremental knowledge transfer strategy effectively alleviates catastrophic forgetting in incremental object detection.

## F. LLM-Based Semantic Concept Construction

**Discriminative Concepts Construction.** As summarized in Algorithm 1, we construct discriminative concepts via LLM-assisted semantic generation. Specifically, for a set of known classes, we enumerate all unordered class pairs and query an LLM with a fixed prompt to obtain one most discriminative visual attribute for each pair. Specifically, we query the LLM with the discriminative prompt $\mathcal{P}_{dis}(c_i, c_j)$: *"Between the following two object classes, identify ONE most discriminative visual attribute that strictly differentiates them."* When new classes are introduced, the discriminative concept set is incrementally extended by applying the same pairwise procedure between newly introduced and previously learned classes, while keeping existing discriminative concepts unchanged.

**Shared Concepts Construction.**

As summarized in Algorithm 2, we construct LLM-derived shared concepts using a two-step reverse-query process. Given a set of known classes, we first enumerate all unordered class pairs and query the LLM with the pairwise prompt $\mathcal{P}_{\text{pair}}(c_i, c_j)$:

---

**Algorithm 1** Discriminative Concepts Construction

---

1: **Input:** known classes at task $t$: $\mathcal{K}_t$; previous known classes $\mathcal{K}_{t-1}$ (empty if $t=1$); LLM prompt template $\mathcal{P}_{dis}(\cdot, \cdot)$
2: **Output:** discriminative concept set $\mathcal{C}_t^u$; pair-to-attribute map $\mathcal{M}_t$
3: $\mathcal{C}_t^u \leftarrow \mathcal{C}_{t-1}^u$, $\mathcal{M}_t \leftarrow \mathcal{M}_{t-1}$ *(if $t=1$, initialize as empty)*
4: $\mathcal{P}_{\text{pair}} \leftarrow \{(c_i, c_j) \mid c_i, c_j \in \mathcal{K}_t,\ i < j\}$ *(all pairs)*
5: **if** $t > 1$ **then**
6: $\quad \mathcal{P}_{\text{pair}} \leftarrow \mathcal{P}_{\text{pair}} \cup \{(c_n, c_o) \mid c_n \in \mathcal{K}_t,\ c_o \in \mathcal{K}_{t-1}\}$ *(new vs. old)*
7: **end if**
8: **for** each $(c_i, c_j) \in \mathcal{P}_{\text{pair}}$ **do**
9: $\quad a_{ij} \leftarrow \text{LLM}(\mathcal{P}(c_i, c_j))$
10: $\quad \mathcal{M}_t[(c_i, c_j)] \leftarrow a_{ij}$
11: $\quad \mathcal{C}_t^u \leftarrow \mathcal{C}_t^u \cup \{a_{ij}\}$
12: **end for**
13: **return** $\mathcal{C}_t^u$, $\mathcal{M}_t$

---

---

**Algorithm 2** Shared Concepts Construction (LLM-Derived)

---

1: **Input:** known classes at task $t$: $\mathcal{K}_t$; prompts $\mathcal{P}_{\text{pair}}(\cdot, \cdot)$ and $\mathcal{P}_{\text{inv}}(\cdot)$
2: **Output:** shared concept set $\mathcal{C}_t^v$; attribute-to-classes map $\Phi_t$
3: $\mathcal{C}_t^v \leftarrow \mathcal{C}_{t-1}^v$, $\Phi_t \leftarrow \Phi_{t-1}$ *(if $t=1$, initialize as empty)*
4: $\mathcal{A}_{\text{new}} \leftarrow \emptyset$
5: $\mathcal{P}_{\text{pair}} \leftarrow \{(c_i, c_j) \mid c_i, c_j \in \mathcal{K}_t,\ i < j\}$ *(class pairs)*
6: **for** each $(c_i, c_j) \in \mathcal{P}_{\text{pair}}$ **do**
7: $\quad \mathcal{A}_{ij} \leftarrow \text{LLM}(\mathcal{P}_{\text{pair}}(c_i, c_j))$ *(shared attributes)*
8: $\quad \mathcal{A}_{\text{new}} \leftarrow \mathcal{A}_{\text{new}} \cup \mathcal{A}_{ij}$
9: **end for**
10: **for** each attribute $a \in \mathcal{A}_{\text{new}}$ **do**
11: $\quad \mathcal{S}_a \leftarrow \text{LLM}(\mathcal{P}_{\text{inv}}(a))$ *(attribute inversion)*
12: $\quad \Phi_t[a] \leftarrow \mathcal{S}_a \cap \mathcal{K}_t$
13: **end for**
14: $\mathcal{C}_t^v \leftarrow \mathcal{C}_t^v \cup \mathcal{A}_{\text{new}}$
15: **return** $\mathcal{C}_t^v$, $\Phi_t$

---

*"List at least $n$ shared visual attributes that BOTH categories have."* to generate shared visual attributes for each class pair. Since a shared attribute may apply to more than the queried pair, we further perform attribute inversion: for each generated attribute $a$, we query the LLM with the inversion prompt $\mathcal{P}_{\text{inv}}(a)$: *"Which of these classes possess the attribute '$\{a\}$'?"* to identify the set of classes in the current known category set that exhibit this attribute, yielding an attribute-to-classes mapping. When a new task is introduced, newly generated shared attributes and their corresponding mappings are merged into the existing shared concept set.

