# OpenReview forum: "Knowing the Unknown: Interpretable Open-World Object Detection via Concept Decomposition Model"
_ICML.cc/2026/Conference — ICML 2026 regular_

### Official Review · Reviewer_fKkH · 2026-02-24

**Soundness:** 2
**Presentation:** 2
**Significance:** 2
**Originality:** 2
**Overall Recommendation:** 1
**Confidence:** 5

**Summary:**

This paper presents IPOW, an interpretable open-world object-detection framework. At its core lies a novel Concept Decomposition Model (CDM) that factorizes complex visual representations into disentangled, semantically explicit components. IPOW constructs the known-class feature space by exploiting the equiangular tight frame (ETF) geometry and discriminative attributes distilled from large language models. In parallel, it builds a shared concept space with transferable properties by integrating external knowledge and a self-supervised sparse auto-encoder, thereby furnishing a principled basis for discovering previously unseen objects. To further separate foreground from background, the framework incorporates a PCA-based background subspace model; deviations from this subspace are flagged as anomalous via reconstruction errors. A Concept-Guided Rectification (CGR) module is subsequently employed to mitigate category confusion arising from visual similarity. Extensive experiments on multiple benchmarks show that IPOW yields substantial improvements in the recall of unknown objects, confirming its effectiveness for open-world detection tasks.

**Compliance With Llm Reviewing Policy:**

Affirmed.

**Ethical Review Concerns:**

see Final Justification

**Ethical Review Flag:**

Flag this paper for an ethics review.

**Ethics Expertise Needed:**

["Research Integrity Issues (e.g., plagiarism)"]

**Final Justification:**

The authors have failed to provide a convincing rebuttal, and several critical issues remain unresolved in the manuscript:

- **Unfair Comparisons.** As previously noted, the current evaluation protocol removes duplicate images, which is misaligned with established evaluation benchmarks. Despite my explicit request for these specific experiments, the authors failed to provide them in their rebuttal. Given that no retraining is required, conducting these evaluations would take `merely 20 minutes`. Nevertheless, the authors continue to rely on the data presented in the original manuscript to support their claims. Consequently, I have reasonable grounds to suspect that the authors are deliberately overstating their contributions through this flawed evaluation setup. Such a lack of academic integrity fundamentally violates the core principles of our research community.

- **Questionable Empirical Results.** The authors have neglected recent advancements in this field. While the most recent work cited in the manuscript is CROWD (NeurIPS 2025), the authors failed to report or compare against its official results. As I highlighted in my previous comments, an evaluation against these official results is imperative. This process would require a mere `5 minutes` of evaluation time; yet, the authors once again failed to provide this assessment. In the manuscript, the performance of CROWD drops drastically from the officially reported `50.5 to 30.4` (S-OWODB). This raises the serious possibility of an intentional suppression of baseline performance.

- **Limited Novelty.** The orthogonal mapping approach has already been introduced in OrthogonalDet; the authors merely extended this existing method to CROWD without presenting significant methodological innovations.

In light of these critical flaws, particularly the questionable academic integrity and the suspected deliberate suppression of baselines, I firmly believe the current manuscript suffers from fundamental issues.

**Key Questions For Authors:**

see Weaknesses

**Limitations:**

yes

**Strengths And Weaknesses:**

Strengths:

- Exceptional transparency and interpretability. By explicitly factorising each RoI representation into discriminative, shared, and background concepts, the detector makes decisions on the basis of concrete semantic attributes rather than opaque feature vectors.

- Rigorous geometric decoupling. The joint use of the Equiangular Tight Frame (ETF) structure from neural-collapse theory and a PCA-based sub-space projection imposes mathematically strict orthogonality constraints, thereby mitigating feature entanglement at its root.

- Superior confusion suppression. The Concept-Guided Rectification (CGR) module alleviates the long-standing confusion between known classes and visually similar unknowns by verifying the completeness of semantic-concept activations.

Weaknesses:

- Reproducibility. The framework relies on several intricate mathematical derivations, yet only pseudo-code is provided. This level of detail is insufficient for faithful reproduction.

- Incomplete comparative study. The main results table omits several representative OWOD methods—ALLOW, EO-OWOD, Hyp-OW, MEPU-FS, OW-OVD, SGROD, SKDF, KTCN, OVOW, and OW-VAP. These baselines should be added in the main paper or appendix. Furthermore, the CROWD baseline is reported under a “top-100 proposals’’ (unofficial) setting; the official protocol uses a much larger pool of unknowns, and IPOW should be evaluated accordingly.

- positive/negative design in discriminative-concept learning. Positives are scored via cosine similarity while negatives are only repelled through a distance loss, which may systematically push negative prototypes too far. An ablation that keeps only positive prototypes is required to justify this design.

- Hyper-parameter sensitivity in the unknown-object score. IPOW defines the unknown score as the maximum of the background score and the shared-concept activation score—two quantities that reflect different phenomena and consequently require extra tuning. A thorough hyper-parameter sensitivity study, together with a visual analysis of why the two scores diverge, is necessary.

- Computational overhead. The proposed differential-feature construction and unknown-probability estimation introduce non-trivial complexity. Inference latency should be reported and compared with that of vanilla Faster R-CNN.

- Potential unknown. PCA is applied to background features that are unlabelled; hence the resulting basis vectors may inadvertently encode unknown objects, giving rise to additional confusion. The manuscript should discuss remedies for this issue and report an upper-bound experiment in which perfect unknown annotations are used to form a “pure’’ background sub-space.

---

> ### Author Rebuttal · Authors · 2026-03-30
>
> Thanks for the constructive feedback, which enhances the clarity and completeness of our work.
>
> **Q1: Reproducibility.**
>
> **R1:**  We will include detailed derivations and full implementation details in the appendix. Our code is at https://anonymous.4open.science/r/IPOW-DD55 and will be released upon acceptance.
>
> **Q2: Incomplete comparison.**
>
> **R2:** These methods most rely on pre-trained detection models which may bring implicit data leakage, making comparisons unfair. We report detailed comparisons and further adapt our method using the same OVD model for fairness.
>
> The results are at: https://anonymous.4open.science/r/IPOW-DD55/assets/MOWOD_compare.png. **Even without pre-trained models, IPOW already beats many methods that rely on pre-training. When equipped with the OVD model, IPOW++ outperforms all the comparisons.**
>
> Under official CROWD setting, IPOW also outperforms CROWD.
>
> | Method | Task 1 U-Recall | Task 2 U-Recall | Task 3 U-Recall |
> | :----: | :-------------: | :-------------: | :-------------: |
> | CROWD  |      57.9       |      53.6       |      69.6       |
> |  IPOW  |      62.4       |      57.3       |      72.4       |
>
> **Q3: Positive/negative design in Disc. concept**
>
> **R3:** Discriminative concepts capture the most distinguishing attributes between category pairs, encouraging known classes to form an equiangular structure. Positive and negative concepts are defined relatively. For example, between car and bicycle, “four wheels” and “two wheels” are discriminative concepts. For car, “four wheels” is a positive concept (pulled closer), while “two wheels” is negative (pushed away); for bicycle, the roles are reversed. Thus, negative concepts are not pushed arbitrarily far, but guided toward the counterpart category.
>
> |          Method           | Task 1 mAP | Task 2 mAP | Task 3 mAP | Task 4 mAP |
> | :-----------------------: | :--------: | :--------: | :--------: | :--------: |
> |     Positive concept      |    59.4    |    46.7    |    39.6    |    36.1    |
> | Positive/negative concept |    62.4    |    52.7    |    45.0    |    42.2    |
>
> Using only positive concepts will hamper the mAP, as it cannot sufficiently separate known classes.
>
> **Q4: Unknown Score Analysis**
>
> **R4:**
>
> 1) The shared-concept score is obtained via a sigmoid, and the background score from normalized reconstruction error; both lie in [0,1]. Max operation is a natural choice for aggregatting the prominent score similar as the role of max pooling.
> 2) We explored different aggregation strategies, for example, by introducing an $\alpha$ that controls the balance between the two components, i.e.
>
> $$
> S_{\mathrm{unk}} = \left( \alpha \cdot S_{\mathrm{unk}}^{\mathrm{share}} + (1 - \alpha) \cdot S_{\mathrm{unk}}^{\mathrm{bg}} \right) \cdot \left( 1 - \max_{j} S_{\mathrm{known}}^{j} \right)
> $$
>
> We study various $\alpha$ on M-OWODB U-Recall, and compare them with the max operation:
>
> |       | $\alpha$ = 0.3 | 0.6  | 0.7      | 0.8  | 0.9  | Max      |
> | :---- | :------------- | :--- | :------- | :--- | :--- | -------- |
> | Task1 | 34.2           | 49.8 | **50.8** | 49.1 | 46.1 | 50.1     |
> | Task2 | 29.4           | 39.6 | 40.6     | 41.0 | 38.2 | **41.9** |
> | Task3 | 31.2           | 44.1 | 46.1     | 45.9 | 43.5 | **46.3** |
>
> It shows that shared concepts dominate, while the background score is complementary. Moreover, the max operation achieves the best performance in most cases without hyper-parameter tuning, demonstrating strong robustness consistent with the intuition of max pooling.
>
> 3) We sample data from the M-OWODB and scatter plot the background and shared-concept scores: https://anonymous.4open.science/r/IPOW-DD55/assets/concept_distribution.pdf. It shows a clear diverge between the two scores. In lower-right part, where  shared-concept scores <  background scores, it indicates few shared attributes with known classes—undetectable by shared concepts alone, but identifiable via background. Table 3 in our paper showed that background concepts greatly improves U-Recall.
>
> **Q5: Computational overhead.**
>
> Please refer to R.cmSn Q2, IPOW adds negligible overhead.
>
> **Q6: Potential unknown.**
>
> **R6:** We conduct an upper-bound experiment on M-OWODB: we use GT to filter out pure background RoIs and perform PCA to construct background concepts, For a controlled comparison, only background concepts are used for unknown prediction in all experiments.
>
> |                 | Task 1 U-Recall | Task 2 U-Recall | Task 3 U-Recall |
> | --------------- | --------------- | --------------- | --------------- |
> | Pure Background | 34.1            | 27.6            | 30.1            |
> | IPOW            | 32.3            | 26.9            | 28.9            |
>
> As shown above, using pure background features brings limited gains. Although unknown objects may be included in the background features, they constitute only a small fraction of background features. So the PCA principal components remain dominated by true background information.

---

> > ### Author Rebuttal · Reviewer_fKkH · 2026-04-02
> >
> > Dear Authors,
> >
> > After carefully reading your rebuttal, I have identified a critical flaw in your experimental setup.
> >
> > Upon reviewing the code provided in your response, it is evident that your codebase inherits the implementation of YOLO-UniOW. However, this introduces a significant problem: YOLO-UniOW adopts the evaluation pipeline of OVOW, which employs a duplicate filtering mechanism during evaluation, specifically via the code mapping[int(imagename)] = imagename (at line 389 in ow_metric.py).
> >
> > This mechanism leads to a significant, artificial inflation of the performance metrics on the M-OWODB benchmark. It is worth noting that this exact issue has already been reported and discussed in the official OVOW GitHub repository (please refer to their GitHub issues). This data processing anomaly constitutes a severe evaluation error.
> >
> > Furthermore, although applying this filtering does not alter the metrics on the S-OWODB benchmark, the performance of your proposed method still fails to surpass that of MEPU-SS (which utilizes Selective Search for modeling).
> >
> > Given these severe evaluation issues, I firmly believe that once the data processing is corrected and the results are legitimately recalculated, the central claims made in your manuscript may no longer hold. I strongly urge you to rigorously re-evaluate your experiments during the subsequent discussion phase and revise your claims and content accordingly.

---

> > > ### Author Response · Authors · 2026-04-02
> > >
> > > **To your final justification (3 to 1):**
> > >
> > > **We have never seen a reviewer with such an arrogant and a biased position to review a scientific paper.**
> > >
> > > **1)Regarding M-OWODB:**
> > >
> > >   **a)Your so-called established benchmark on M-OWODB is flawed and apparently you know it.** Your position is biased that the community should follow an incorrect protocol, which is already realized by the community and all rigorous scholars have tried their best to correct the flawed evaluation, such as the accepted papers RandBox (ICCV 2023), OrthogonalDet (CVPR 2024), and CROWD (NeurIPS 2025) etc.
> > >
> > >   **b)We have reported or re-evaluated all the comparison methods on the corrected M-OWODB for fairness, you seemed overlooked our effort and force us to report results on a flawed protocol.** Following a flawed protocol rather than evaluating existing methods on a corrected protocol is what you called academic integrity? All the reported results including the baselines in our paper and the rebuttal can be reproduced and we are willing to release our code. We can not understand why and how you suspected we deliberately suppress the baselines?
> > >
> > >   **c)The experimental results during the rebuttal and discussion period took us "near two weeks" to re-train every comparison method, as most of authors do not release their trained models, far from the "20-minute task" you casually asserted.**
> > >
> > >   **d)Even on the flawed protocol, our method can still outperform all the comparison methods in our Table 1.** But we think that is not a correct way to promote the development of this community. It will lead to wrong and wrong evaluations in the future.
> > >
> > > **2)Comparison with CROWD:**
> > > **We have already provided a direct comparison with CROWD in our rebuttal (R2), strictly following the official evaluation protocol of CROWD.** It is regrettable that you apparently overlooked this and raised an intentional suppression of baseline?
> > >
> > > **3)Limited novelty?** Are you serious that we extend OrthogonalDet+CROWD?
> > >
> > > **A rigorous reviewer should evaluate a paper based on facts rather than personal subjective biases. A failure to comply with your unreasonable suggestions cannot serve as a basis for you to question academic integrity.**
> > >
> > >
> > > ----------------------------
> > >
> > > **Dear Reviewer,**
> > >
> > > **1)First, our evaluation metrics and claims are correct and compelling.**
> > >
> > > 2)The issue you raised is because there are duplicate images in M-OWODB test set, and the code performs deduplication for true performance reflection. This is not specific to OVOW or YOLO-UniOW, but rather a line of methods in the OWOD community all adopted the same evaluation strategy, such as RandBox, OrthogonalDet, and CROWD.
> > >
> > > 3)The original M-OWODB test set contains 11.6% duplicated images, and the community already realize this issue and is trying to correct it.  S-OWODB, proposed by OW-DETR has already removed duplicate images, RandBox eliminates duplicate images via the test.json file. OrthogonalDet modifies the evaluation code to deduplicate the M-OWODB test set.
> > >
> > > **4) In that GitHub issue discussion, the poster explicitly acknowledges the problem (“there are duplicate filenames, amounting to as many as 1127.”), yet he/she insists on continuing with the flawed evaluation.** His/Her position is that the community should follow an incorrect protocol— a stance that we cannot endorse because it compromises scientific rigor.
> > >
> > > **5)The main baselines we compare against, such as RandBox, OrthogonalDet, and CROWD, all follow our same evaluation protocol.** Some earlier methods which do not perform such deduplication, we re-evaluated them on the M-OWODB test set for fairness; the results are shown below. On the SOWOD test set, where no duplicate images exist, the results can be referred to in our Table 1.
> > >
> > > | Methods       | T1UR | T1mAP | T2UR | T2mAP | T3UR | T3mAP | T4mAP |
> > > | :------------ | :--- | :---- | :--- | :---- | :--- | :---- | :---- |
> > > | ORE           | 12.5 | 60.4  | 9.64 | 41.8  | 10.4 | 33.1  | 30.3  |
> > > | OW-DETR       | 13.1 | 64.6  | 6.28 | 47.9  | 8.12 | 42.1  | 37.8  |
> > > | PROB          | 26.2 | 66.7  | 24.2 | 50.8  | 25.4 | 44.4  | 39.8  |
> > > | CAT           | 29.8 | 66.9  | 25.8 | 51.2  | 30.2 | 42.2  | 37.9  |
> > > | RandBox       | 10.6 | 61.8  | 6.3  | 45.3  | 7.8  | 39.4  | 35.4  |
> > > | OrthogonalDet | 24.6 | 61.3  | 26.3 | 47.0  | 29.1 | 41.3  | 37.9  |
> > > | CROWD*        | 42.9 | 61.7  | 31.4 | 47.8  | 34.7 | 42.5  | 38.5  |
> > > | **IPOW**      | 50.1 | 62.4  | 41.9 | 52.7  | 46.3 | 45.0  | 42.2  |
> > >
> > > 6)Regarding the results on S-OWODB, our method does not surpass MEPU-SS. We have explained in the rebuttal: it leverages a pre-trained model SoCo, making the comparison unfair. Without introducing pre-trained detection models, our method achieves SOTA definitely.
> > >
> > > 7)Our main claim has been validated through extensive experiments. We argue that the claims firmly hold **whether duplicate images exist or not**.
> > >
> > > **Our evaluation is correct, our main comparisons are consistent and fair.**
> > >
> > > The authors.

---

### Official Review · Reviewer_DyeL · 2026-02-26

**Soundness:** 4
**Presentation:** 4
**Significance:** 4
**Originality:** 4
**Overall Recommendation:** 5
**Confidence:** 5

**Summary:**

The authors propose a concept-driven interpretable open-world object detection framework. They introduce a concept decomposition model (CDM) to explicitly decompose the features of the region of interest (RoI) into a discriminative concept space, a shared concept space, and a context concept space.

**Compliance With Llm Reviewing Policy:**

Affirmed.

**Final Justification:**

My concerns have been fully addressed, so I will keep my positive evaluation.

**Key Questions For Authors:**

As I mentioned in Weakness, I hope the authors can provide more visualizations, especially in complex scenarios, to demonstrate the true performance of the model in recognizing unknown objects.

**Limitations:**

As the number of tasks increases, the discriminative concept pairs grow. It would be beneficial to discuss the scalability of the concept dictionary and whether the residual shared concepts might eventually become redundant or noisy in very large-scale open-world settings.

**Strengths And Weaknesses:**

Strength:

1. This paper analyzes the OWOD task from the perspective of Concept Decomposition and proposes a novel OWOD framework.

2. This method achieves state-of-the-art (SOTA) performance on most existing benchmarks (M-OWODB, S-OWODB, and so on), showing a significant improvement in U-Recall while maintaining a high performance for known categories.

3. The paper is clearly structured and easy to read.

Weakness:

1. This paper mainly uses U-Recall to measure the performance of unknown objects. However, recall does not show the "accuracy" or precision of detection results. High recall often comes at the cost of huge false positives. The author should provide more comprehensive results to demonstrate the accuracy of these unknown object bboxs.

2. Figures 5 and 6 mainly show relatively clear scenarios and objects. To further validate the model's performance for unknown objects, the authors should provide more visualizations of complex environments. Especially,  the unknown targets have a high degree of visual similarity with the background environment or known objects.

---

> ### Author Rebuttal · Authors · 2026-03-30
>
> Thanks for recognizing our work and for the constructive suggestions.
>
> **Q1: Accuracy evaluation of unknown detection**
>
> **R1:** Thanks for the suggestion. The accuracy of unknown objects is indeed a major challenge in current OWOD methods. Without pre-training knowledge, it is difficult to accurately detect unknown categories solely based on learning from known classes. This is also why our method and prior works do not report accuracy for unknown objects. However, this does not lead to huge false positives, as OWOD limits the number of proposals per image (e.g., at most 100 proposals per image in our evaluation protocol).
>
> OWOD was originally proposed to focus on two core metrics: Unknown Recall (U-Recall) and the confusion between known and unknown classes (measured by metrics WI and A-OSE, where WI measures the drop in known-class precision caused by unknown instances, and A-OSE counts the number of unknown objects misclassified as known classes, thus reflecting their confusion).
>
> Although OWOD does not enforce accuracy for unknown objects, this does not affect its practicality. OWOD is an incremental learning process, where the unknown objects predicted at each stage can be further identified for categories of interest and then annotated, and subsequently learned in the next stage, allowing the model to continuously evolve.
>
> **Q2: More visualizations of complex environments**
>
> **R2:** To further validate our model’s performance on challenging unknown objects, we evaluate and visualize IPOW in more complex environments. The images are from the Camouflaged Object Detection dataset COD10K, which contains complex scenes where targets have high visual similarity with the background. The visualization results are available at: https://anonymous.4open.science/r/IPOW-DD55/assets/vis_complex_environments.pdf.
>
> As shown in the figure, complex environments pose challenges for IPOW. However, since our method detects unknown objects through shared attributes, even in complex backgrounds, if some attribute features of the unknown object are exposed, IPOW can detect it based on these attributes. For example, for a cat hidden in the grass, the model can detect it through its exposed ears. This demonstrates the robustness of our method. We will include these additional visualizations in the next version of the paper.
>
> **Limitations**
>
> As the number of tasks increases, for discriminative concepts,  discriminative concept pairs can overlap  in practice (e.g., “four legs vs. two legs” for dog–human and horse–human, while dog–horse may use “large vs. small body size”), which does not affect the separability of known classes. As for shared concepts, as more known categories are learned, the concept space tends to saturate, and adding new classes may not introduce additional shared concepts. We will further discuss the scalability in the revised version.

---

> > ### Author Rebuttal · Reviewer_DyeL · 2026-04-04
> >
> > The author has fully addressed my concerns.

---

> > > ### Author Response · Authors · 2026-04-05
> > >
> > > Thanks again for the reviewer's valuable suggestions.

---

### Official Review · Reviewer_KSqR · 2026-03-08

**Soundness:** 3
**Presentation:** 3
**Significance:** 3
**Originality:** 3
**Overall Recommendation:** 5
**Confidence:** 4

**Summary:**

This paper proposes an interpretable open-world object detection framework based on concept decomposition, aiming to address known–unknown confusion by introducing discriminative, shared, and background concept spaces together with a concept-guided rectification mechanism.

**Compliance With Llm Reviewing Policy:**

Affirmed.

**Final Justification:**

Please refer to my response in the acknowledgement.

**Key Questions For Authors:**

See `Weaknesses' above.

**Limitations:**

The authors are encouraged to briefly address model limitations (e.g., reliance on pretrained LLM/CLIP and potential bias) and possible risks in real-world deployment.

**Strengths And Weaknesses:**

Strengths:

1. The paper tackles an important yet underexplored aspect of OWOD: interpretability at the concept level rather than purely improving unknown recall.
2. The decomposition into discriminative, shared, and background concepts is well-motivated and theoretically grounded (e.g., Neural Collapse).
3. The experiments also show the strong performance compared with existing methods.

Weaknesses:

1. The method leverages LLM-derived concepts and CLIP text embeddings. It is unclear whether the comparisons with prior OWOD methods are fully fair, especially if those methods do not use additional language supervision or pretrained vision-language models. A clearer discussion on fairness and experimental protocol would be helpful.

2. The framework is built upon Faster R-CNN, which is relatively dated. It would strengthen the paper to either:

    a. Provide results with more recent detectors (e.g., DETR and its variants), or

    b. Justify more clearly why Faster R-CNN is chosen and how the method generalizes to other backbones.

    c. In addition, since CLIP is already introduced for Open-Vocabulary Detection, why the detection backbone is not built directly on CLIP-style visual encoders.

3. The distinction between the “attributes” used in Section 4.3 (discriminative concepts) and those in Section 4.4 (shared concepts) is not sufficiently clear. Both appear to describe semantic attributes. A more explicit comparison of their roles, construction, and supervision signals would improve clarity.

4. he training and inference procedures require clearer explanation. In particular:

    a. How are known and unknown classes handled during training?

    b. How are unknown classes defined in practice? Are they predefined splits, or simply any object not belonging to known classes?

    c. During inference, how is the final decision between known and unknown made step by step?

5. The illustration of discriminative concepts in Figure 2 is somewhat confusing.

---

> ### Author Rebuttal · Authors · 2026-03-30
>
> Thanks for the thoughtful feedback, which helps us clarify and improve our work.
>
> **Q1: Comparison Fairness.**
>
> **R1:** LLM is strictly limited to summarizing shared or discriminative attributes between two classes. For CLIP, we only use its text encoder to encode these LLM-generated attributes, serving purely as a text encoding tool. Any additional pretraining is solely at the textual level. The core challenge of OWOD is detecting unknown classes; we do not obtain any information about unknown classes from the textual side, nor introduce any prior knowledge about them on the visual side. Therefore, the difficulty of detecting unknowns is the same as for other methods and the comparisons is fair.
>
> **Q2: Model architecture choice**
>
> **R2:**  **a&b:** As stated in our paper, our method performs concept decomposition on RoI-level features, since RoIs fully encode the region-level information of potential objects. This property is essential for our design which also motivates us to choose FasterRCNN naturally. For DETR-style models, each detection originates from an object query. Although the concept decomposition can be applied to these queries, we choose the extention as a future work as it is non easy to equip the CBM into DETR in 1-2 weeks. And in this work, we investigate the effectiveness of concept-based regional feature decomposition for mining unknowns and distinguishing known from unknown, while enhancing the interpretability of the OWOD framework.
>
> **c:** For CLIP-style visual encoders, their strong zero-shot capability enables effective encoding of unseen categories. However, this may leads to two issues: 1) the pretraining knowledge may result in unfair comparisons, and 2) it may partially violate the open-world setting, as unknown objects may have already been seen during pretraining.
>
> **Q3: Disc. concepts v.s. shared concepts**
>
> **R3:**
>
> - Roles: Discriminative concepts are the most distinguishing attributes between pairs of classes, designed to maximize the separation between known categories for accurate classification. In contrast, shared concepts are common attributes shared across multiple categories. Their role is to capture semantic commonalities among known classes, enabling the model to generalize and detect unknown objects.
> - Construction: Discriminative concepts are summarized by an LLM to identify the most contrasting attributes strictly between Class A and Class B. For shared concepts, to ensure a complete and robust concept space, we first use an LLM for initial summarization and then employ a Sparse Autoencoding mechanism to discover residual shared concepts that fall outside the scope of the LLM’s summaries.
> - Supervision Signals: Shared concepts are optimized via BCE loss, requiring that a target of a specific class activates all its corresponding shared concepts. Discriminative concepts are supervised by a margin-based contrastive loss, which pulls the feature representation closer to the positive concepts of that category while pushing away the negative ones.
>
> **Q4: Training and inference details**
>
> **R4:**
>
> a. During training, known classes are annotated while unknown classes are not. For known classes, we mine discriminative concepts for classification and shared concepts for unknown detection. Since unknown categories have no labels during training, there is no specific processing or supervision explicitly directed at them at this stage.
>
> b. In the OWOD setting, unknown classes follow predefined splits. As the OWOD process is inherently an incremental learning task, unknown classes are defined as categories that will be learned in future stages. For example, in MOWOD benchmark, the known classes for Task 1 are the 20 classes from VOC, while the unknowns are the 60 classes from the COCO dataset to be learned in subsequent tasks. In the test set, these future categories are collectively labeled as "unknown." for evaluation.
>
> c. IPOW first predicts potential unknown through shared and background concepts, while predicting known via discriminative concepts. To resolve known-unknown confusion, we apply Concept Guided Rectification as described in Section 4.6 to adjust the prediction scores. The core logic is that a known class object must demonstrate "full activation" of its predefined shared semantic concepts. Using this insight, we rectify the scores for known classes; the final unknown are then determined by removing these rectified known objects from the initial pool of potential unknown.
>
> **Q5: Confusing Fig.2**
>
> **R5:** As illustrated in Figure 2, where the blue circles represent class prototypes, we construct discriminative concepts for every pair of categories. For instance, considering the prototypes for "car" and "bicycle," the discriminative concepts are "four wheels" and "two wheels," which form a positive-negative concept pair. For the "car" class, "four wheels" serves as the positive concept while "two wheels" is the negative concept. We will improve Fig.2.

---

> > ### Author Rebuttal · Reviewer_KSqR · 2026-04-01
> >
> > The author has fully addressed my concerns. Therefore, I will raise my score to accept.

---

> > > ### Author Response · Authors · 2026-04-01
> > >
> > > Thanks again for the reviewer's valuable suggestions.

---

### Official Review · Reviewer_cmSn · 2026-03-11

**Soundness:** 3
**Presentation:** 3
**Significance:** 3
**Originality:** 3
**Overall Recommendation:** 4
**Confidence:** 4

**Summary:**

This paper addresses a critical gap in open-world object detection (OWOD): the lack of interpretability in existing methods, which leads to persistent known–unknown confusion and low unknown recall. The authors propose a concept-driven interpretable framework (IPOW) built on a Concept Decomposition Model (CDM) that breaks down RoI features into discriminative, shared, and background concepts, and introduce a Concept-Guided Rectification (CGR) module to resolve misclassification caused by unknown objects falling into the discriminative space of known classes. Extensive experiments on multiple benchmarks (M-OWODB, S-OWODB, DIOR, PASCAL VOC 2007) demonstrate state-of-the-art performance in both unknown recall and known-class mAP, while also delivering concept-level interpretability for all predictions. Overall, this work makes a valuable contribution to OWOD by merging performance improvements with interpretability, a combination that is rare in current research.

**Compliance With Llm Reviewing Policy:**

Affirmed.

**Final Justification:**

The author solved my problem, so I will maintain my original positive score.

**Key Questions For Authors:**

Please refer to the weakness section.

**Limitations:**

A detailed analysis of the specific causes underlying the performance degradation observed in both comparative and ablation experiments is essential, followed by a thorough discussion of the methodological limitations inherent to the proposed approach.

**Strengths And Weaknesses:**

### Strengths:
Novel Interpretable Framework Design: The core CDM decomposition is a well-motivated innovation that rethinks OWOD from a concept modeling perspective. By splitting RoI features into three semantically meaningful concepts, the framework moves beyond the "black box" design of existing OWOD methods (e.g., class-agnostic objectness heads, random region proposals) and provides explicit reasoning for both known and unknown predictions. This interpretability is not just a secondary feature but a foundational element that drives the solution to known–unknown confusion.

### Weakness:
1.	The paper mentions using an LLM to generate discriminative and shared concepts but provides no details on the specific LLM (e.g., GPT-4, Llama) or prompt engineering strategies beyond high-level templates. Different LLMs or prompt designs may yield varying concept quality, which could impact IPOW’s performance. The authors should add a study on the sensitivity of results to LLM choice/prompting.
2.	Computational Overhead Analysis: The paper notes that the LLM incurs no additional overhead, but it does not report the overall computational cost (e.g., inference time, FLOPs) of IPOW compared to baseline OWOD methods (e.g., CROWD, OrthogonalDet).
3.	The paper briefly criticizes OVD-based OWOD methods for implicit data leakage but does not include direct quantitative comparisons with state-of-the-art OVD methods (e.g., OW-OVd). While OVD and OWOD have different formulations, a head-to-head comparison would better contextualize IPOW’s performance, especially on benchmarks where OVD methods are strong.
4.	The paper shows concept activation score distributions (Fig. 3) but does not provide direct visualization of concept activations on images (e.g., heatmaps of discriminative/shared concept activation for a given RoI). Adding such visualizations would further strengthen the interpretability claim.

---

> ### Author Rebuttal · Authors · 2026-03-30
>
> We sincerely thank the reviewer for the positive feedback and constructive suggestions.
>
> **Q1: Lack of clarity and analysis on LLM usage.**
>
> **R1:** In our method, the LLM is only used to summarize concepts based on category names, which does not require strong generative capabilities. Therefore, the choice of LLM has limited impact on our approach. For prompt design, since **commonly used prompts** (e.g., “list shared attributes across categories”) often produce sparse and inaccurate concepts, we adopt an **inversion prompt strategy** as described in Appendix F (Shared Concepts Construction) . We thus design the inversion prompt to address this issue.
>
> | **LLM Model** | Prompt Strategy      | U-Recall | mAP  | WI     | A-OSE |
> | ------------- | -------------------- | -------- | ---- | ------ | ----- |
> | GPT-4         | Inversion prompt     | 50.1     | 62.4 | 0.0369 | 3648  |
> | LLaMA-2       | Inversion prompt     | 49.8     | 62.1 | 0.0377 | 3652  |
> | GPT-4         | Commonly used prompt | 47.2     | 61.9 | 0.0388 | 3713  |
> | LLaMA-2       | Commonly used prompt | 47.9     | 62.0 | 0.0373 | 3689  |
>
> As shown in the table (M-OWODB T1), the choice of LLM has little impact on performance, while the prompt design does affect the results (U-Recall +2.9 for GPT-4 and +1.9 for LLaMA-2). Our inversion prompt consistently achieves better performance.
>
> **Q2: Computational Overhead Analysis**
>
> **R2:** We compare computational cost in terms of FLOPs, inference speed, and parameters. IPOW corresponds to the Task 4 model in MOWOD, where complexity is highest. Inference speed is measured on a single GPU without hardware acceleration.
>
> | Method                 | FLOPs      | inference time | Params    |
> | ---------------------- | ---------- | -------------- | --------- |
> | Faster-rcnn(base line) | 184 GFLOPs | 25.16 FPS      | 41.446M   |
> | OrthogonalDet          | —          | 1.65 FPS       | 105.994M  |
> | CROWD                  | —          | 1.68 FPS       | 106.216 M |
> | IPOW                   | 185 GFLOPs | 19.91FPS       | 43.734M   |
>
> As shown in the table, IPOW introduces negligible overhead compared to the Faster R-CNN baseline, with only a minor decrease in inference speed. In contrast, prior OWOD methods such as OrthogonalDet and CROWD suffer from substantial inference slowdown, as they rely on prior methods (RandBox) that require iterative refinement of randomly generated proposals. This also motivates our design of the GMM RPN to replace such approaches.
>
> **Q3: Comparison with OVD-based methods**
>
> **R3:** Indeed, a direct comparison with OVD-based methods can provide a better understanding of IPOW’s performance. We provide detailed comparisons on M-OWODB as follows. To ensure a fair comparison, we further reproduce our method using the same OVD model (YOLO-World), denoted as IPOW++.
>
> |            |              |    Task 1    |         |    Task 2    |         |    Task 3    |         | Task 4  |
> | :--------: | :----------: | :----------: | :-----: | :----------: | :-----: | :----------: | :-----: | :-----: |
> | **Method** | **Pretrain** | **U-Recall** | **mAP** | **U-Recall** | **mAP** | **U-Recall** | **mAP** | **mAP** |
> |    SKDF    |  YOLO World  |      39      |  56.8   |     36.7     |  40.3   |     36.1     |  30.1   |  26.9   |
> |   OW-OVD   |  YOLO World  |      50      |  69.4   |     51.7     |  55.6   |     50.6     |   47    |  41.6   |
> |   OW-VAP   |  YOLO World  |     58.8     |  68.8   |     56.3     |  55.6   |     55.1     |  47.1   |   42    |
> |    IPOW    |      -       |     50.1     |  62.4   |     41.9     |  52.7   |     46.3     |   45    |  42.2   |
> |  IPOW ++   |  YOLO World  |     **60.2**     |  **69.6**   |     **59.8**     |  **56.4**   |     **60.9**    |   **48**    |  **42.3**   |
>
> As shown above, IPOW achieves competitive performance without any pre-trained models and outperforms several methods that rely on pre-training. When using the same OVD model, IPOW++ achieves state-of-the-art performance across multiple tasks.
>
> **Q4: Lack of visualization of concept activations**
>
> **R4:** We provide visualization results of concept activations (discriminative and shared concepts on RoIs) at: https://anonymous.4open.science/r/IPOW-DD55/assets/roi_concept_feature.pdf. These will be included in the revised version.
>
> As shown in Fig. (c), the known class “dog” strongly activates its corresponding discriminative concepts (e.g., “fur”, “four legs”). However, as shown in Fig. (a), since discriminative concepts capture only the most distinguishing features among known classes, unknown objects may also activate these concepts, leading to confusion and misclassification as known classes. In contrast, as shown in Fig. (d), known categories fully activate their corresponding shared concepts, while, as shown in Fig. (b), unknown objects may only partially activate shared concepts. Based on this observation, we design the Concept-Guided Rectification module to further correct such confusion.

---

> > ### Author Rebuttal · Reviewer_cmSn · 2026-04-03
> >
> > Thank you for the detailed and thoughtful rebuttal. My concerns have been fully addressed, so I will keep my positive evaluation.

---

> > > ### Author Response · Authors · 2026-04-03
> > >
> > > We sincerely thank the reviewer again for the valuable suggestions.

---

### Decision · Program_Chairs · 2026-04-30

**Decision:**

Accept (regular)

**Comment:**

The paper proposes IPOW, an interpretable open-world object detection framework designed to improve unknown-object detection while reducing confusion between known and unknown classes. The method introduces a Concept Decomposition Model (CDM) that decomposes RoI features into discriminative, shared, and background concepts, as well as a Concept-Guided Rectification (CGR) module that addresses cases in which unknown objects are incorrectly mapped into the discriminative space of known classes. Overall, the proposed framework aims to improve unknown recall, mitigate known–unknown confusion, and provide concept-level interpretability for both known and unknown predictions.

Overall, there is broad agreement among the reviewers on the novelty and technical merit of the paper. The rebuttal helped clarify several points that were previously ambiguous. In addition, the extra experiments provided during rebuttal further strengthened the empirical support for the method.

One reviewer raised a valid concern regarding the fairness of the comparison, noting that the use of a duplicate filtering mechanism could lead to inconsistencies relative to the reported baselines. The authors addressed this concern through additional clarification and new experimental results. In my assessment, their response adequately resolves the issue. Importantly, this concern does not materially weaken the main technical contribution of the paper or the overall strength of its empirical findings, which have been generally recognized by the other reviewers as well.

For the final version, the authors are encouraged to describe the experimental protocol more clearly, particularly the evaluation setting and how the reported results are obtained, in order to avoid similar confusion for future readers.